# Mesophotic coral bleaching associated with changes in thermocline depth

**Clara Diaz** [1,2] ✉, **Nicola L. Foster** [1,2] ✉, **Martin J. Attrill** [1], **Adam Bolton**[1], **Peter Ganderton** [1], **Kerry L. Howell**[1], **Edward Robinson** [1] & **Phil Hosegood** [1,2] ✉

As global temperatures continue to rise, shallow coral reef bleaching has become more intense and widespread. Mesophotic coral ecosystems reside in deeper (30–150 m), cooler water and were thought to offer a refuge to shallow-water reefs. Studies now show that mesophotic coral ecosystems instead have limited connectivity with shallow corals but host diverse endemic communities. Given their extensive distribution and high biodiversity, understanding their susceptibility to warming oceans is imperative. In this multidisciplinary study of an atoll in the Chagos Archipelago in the central Indian Ocean, we show evidence of coral bleaching at 90 m, despite the absence of shallow-water bleaching. We also show that the bleaching was associated with sustained thermocline deepening driven by the Indian Ocean Dipole, which might be further enhanced by internal waves whose influence varied at a sub-atoll scale. Our results demonstrate the potential vulnerability of mesophotic coral ecosystems to thermal stress and highlight the need for oceanographic knowledge to predict bleaching susceptibility and heterogeneity.

Coral reefs represent one of the most iconic, diverse and valuable ecosystems on Earth, hosting almost 30% of all marine fish species and possibly more than a million species overall[1,2]. In spite of their limited spatial coverage (0.5% of the marine world), they support more species per unit area than any other marine environment[3]. Living close to their thermal limit, however, shallow corals are threatened by the global increase in sea surface temperatures (SSTs) and subsequent coral bleaching; current estimates suggest that 70–90% of the reefs will be lost with a 1.5 °C rise and 99% if the increase in temperatures reaches 2 °C[4,5]. Bleaching events are increasing in their severity, frequency and duration[6,7], yet our knowledge of the impacts of these events is centred around shallow-water coral reefs. Mesophotic coral ecosystems (MCEs) reside between the depths of 30–150 m and have been estimated to occupy two-thirds of the total depth range of zooxanthellate coral environments[8,9]. It was previously hypothesised that these communities are buffered against anthropogenic changes as they are located in deeper, cooler water, and may therefore offer refuge to shallow-water corals through the dispersal of larvae[8,10]. However,

recent studies have shown that MCEs have limited ecological and genetic connectivity to their shallow-water counterparts and are biologically and ecologically valuable in their own right, hosting diverse communities with high levels of endemism[11–14]. Given the extensive distribution of MCEs and the high biodiversity they support, understanding the susceptibility of these communities to bleaching in the face of warming oceans is imperative.

Mesophotic corals are thought to be buffered from the thermal stress experienced by shallow-water corals as they live in deeper waters and particularly within the thermocline across which temperature decreases abruptly[8,15]. Vertical perturbations in the depth of the thermocline expose benthic fauna and flora to rapid, and pronounced, changes in temperature. Recent results demonstrate that internal waves breaking over submarine slopes lift the thermocline, drawing cold water upwards and relieving thermal stress[16,17]. Rather than the relatively easily monitored SST, however, the dynamical processes responsible for modulating the thermocline depth occur independently of surface warming and are much harder to monitor with in situ sensors. Thus,

[1]School of Biological and Marine Sciences, University of Plymouth, Drake Circus, Plymouth PL4 8AA, UK. [2]These authors contributed equally: Clara Diaz, Nicola L. Foster, Phil Hosegood. ✉e-mail: clara.diaz@plymouth.ac.uk; nicola.foster@plymouth.ac.uk; phil.hosegood@plymouth.ac.uk

whilst the threat of shallow coral bleaching can be assessed, and early warning indicators provided through products such as the National Oceanic Atmospheric Administration (NOAA) Coral Reef Watch (CRW), a metric which nonetheless has to be interpreted with caution)[18,19], the susceptibility of MCEs to thermal stress requires an understanding of the physical oceanographic drivers of temperature variability beneath the surface, which is a considerably more challenging prospect than predictions of large-scale surface temperatures. The problem is further compounded by the practical and technological challenges of monitoring the coral community at mesophotic depths.

Here, we reveal the susceptibility of MCEs to bleaching events at depths of up to 90 m at two sites within an isolated atoll within the Chagos Archipelago in the central Indian Ocean (Fig. 1). This is the deepest example of coral bleaching on record and occurred in the absence of shallow-water coral bleaching. We demonstrate an association with the prevailing physical oceanographic regime in, firstly, triggering a mesophotic coral bleaching event at the basin scale due to a strong Indian Ocean Dipole (IOD) event and, secondly, in potentially modulating the extent of bleaching at this depth around the atoll due to the internal wave field.

Together, our results confirm that bleaching of scleractinian corals can occur within MCEs at depths of up to 90 m and, in the absence of shallow-water bleaching, we show the limitations of using surface bleaching susceptibility indicators to represent the threat to MCEs. Our observations also highlight the critical role played by the physical oceanographic regime at multiple spatiotemporal scales in setting the subsurface thermal regime.

## Results and discussion
### Coral bleaching at depth

We used a combination of remotely operated vehicle (ROV) surveys to obtain high-resolution digital imagery of the coral colonies and their bleaching state from shallow (15 m) to lower mesophotic depths (90 m), moored in-situ measurements of the physical oceanographic regimes over the slope surrounding the atoll to resolve the dynamical processes modulating the thermocline depth, and high-resolution numerical modelling to simulate the change in oceanographic conditions between the two sites surveyed. Publicly available numerical model outputs of conditions across the Indian Ocean were also used to understand the regional scale conditions that reflected the influence of

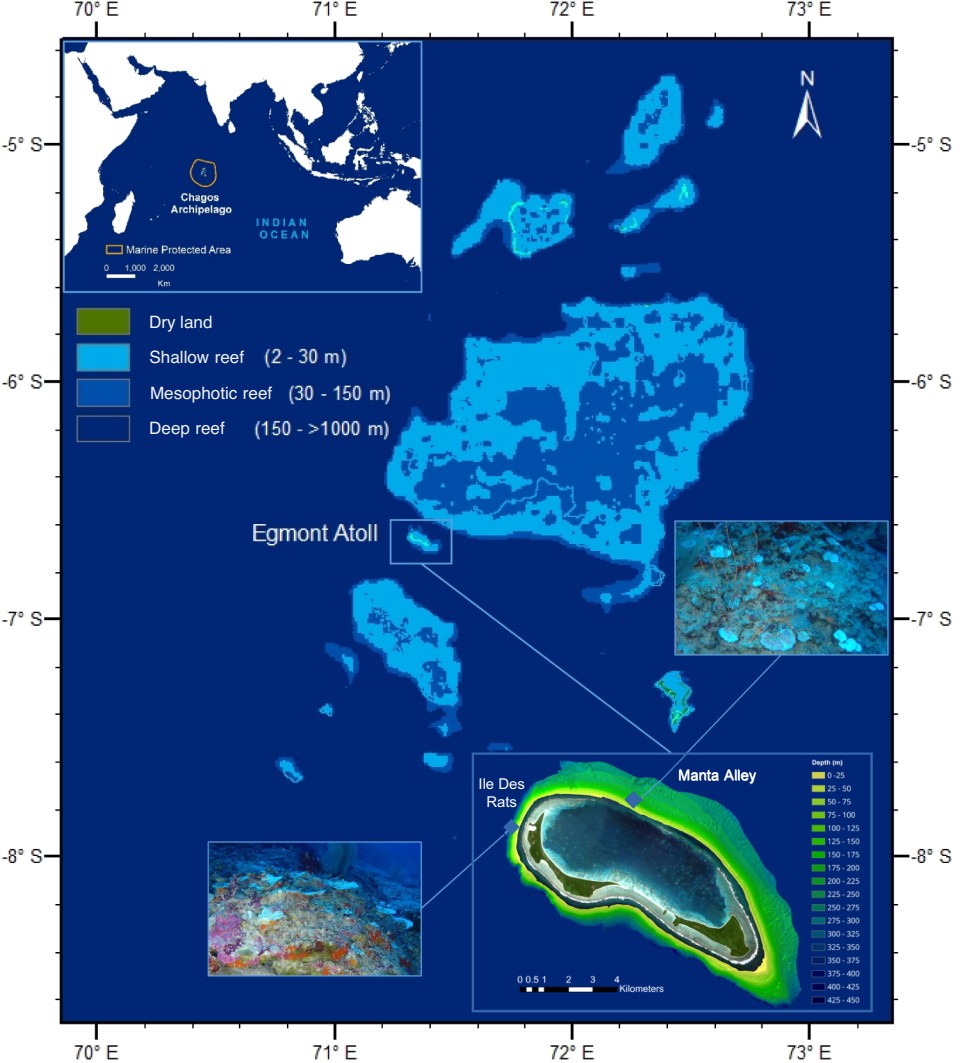

**Fig. 1 | Study sites and their locations.** Composite figure showing Egmont Atoll (right corner inset) located within the Chagos Archipelago, central Indian Ocean (left corner inset), and the Marine Protected Area delimitation. The depth gradients were extracted from GEBCO data and defined the benthos delimitations. Egmont Atoll is displayed using Copernicus Sentinel data 2020, with 3D bathymetry data acquired using a multibeam with the two study sites highlighted: Ile Des Rats (North-western flank) and Manta Alley (North-eastern flank). The two images show examples of bleached corals observed at approximately 65 m for the two respective sites. The original map can be found here: https://en.wikipedia.org/wiki/Chagos_Archipelago#/media/File:Chagos_map.PNG.

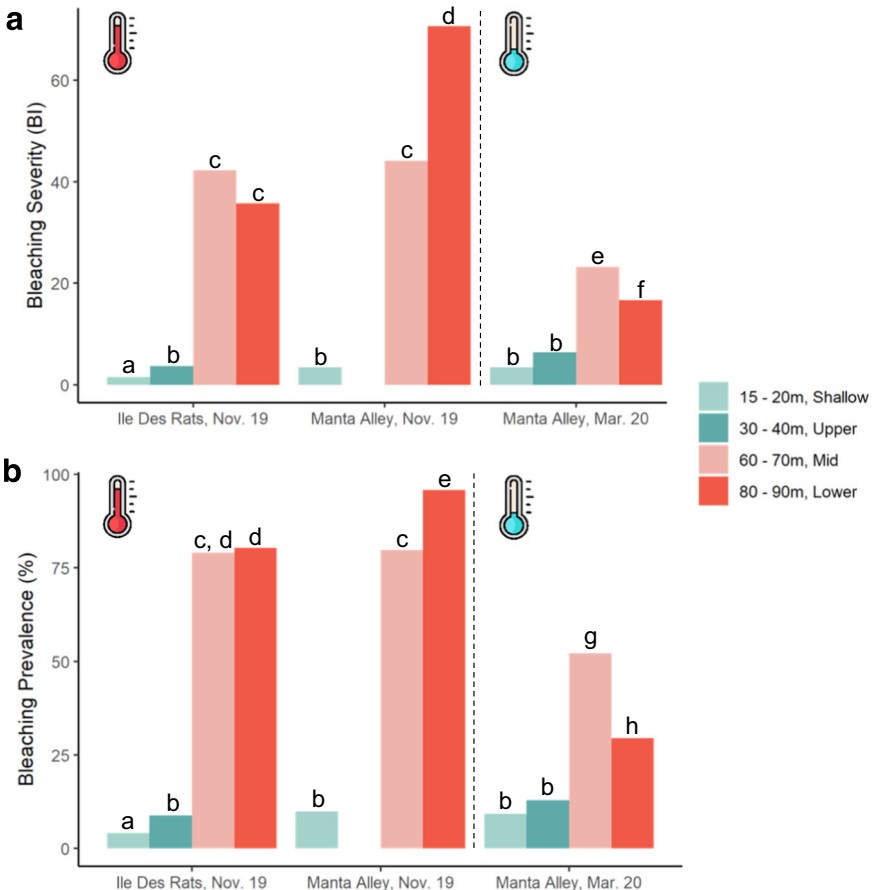

**Fig. 2 | Coral bleaching severity and prevalence at Egmont Atoll in November 2019 and March 2020. a** Bleaching severity (bleaching index, BI) and **b** Bleaching prevalence (percentage, %) at the 2 sites surveyed at Egmont Atoll (Ile Des Rats and Manta Alley) in November 2019 (Nov. 19) and March 2020 (Mar. 20). Different letters = p-values < 0.05 for Kruskal−Wallis tests (see supplementary materials for full details). Each depth band is represented by a different colour and corresponds to a depth zone (shallow water; upper-, mid-, lower-mesophotic zone). The 30–40 m depth band in Manta Alley, Nov.19 was not surveyed due to lack of time and strong currents. The dashed line separates the survey undertaken in November 2019 and March 2020, respectively. The thermometers represent figurative water temperature in mesophotic depths and were adapted from www.flaticon.com.

the IOD. The IOD is a key interannual cycle operating with a periodicity of 3−7 years that modulates surface wind fields, ocean currents and thermocline depth throughout the tropical Indian Ocean in a similar manner as El Niño Southern Oscillation (ENSO) in the Pacific[20].

Video surveys of MCEs were conducted at two contrasting sites within Egmont Atoll, Ile de Rats (IDR) on the North-western flank and Manta Alley (MA), located on the North-eastern side of the atoll (Fig. 1), during November 2019, when the IOD exhibited its strongest positive phase on record, and March 2020 by which time the IOD was weakly negative. Coral bleaching was assessed using two metrics. Firstly, bleaching prevalence was estimated as the percentage of affected individual colonies in the population. Secondly, bleaching severity was estimated as the bleaching extent within a single colony, calculated using a bleaching index (BI).

During November 2019, coral bleaching was virtually absent on shallow-water (15−20 m) and upper-mesophotic reefs (30−40 m), with a BI < 2% and <10% of all colonies bleached at both sites (Fig. 2a, b; full results can be viewed in Supplementary Tables S6 and S7). In contrast, mid- and lower-mesophotic depths (60−70 and 80−90 m, respectively) experienced significant coral bleaching in both severities (BI > 26; Fig. 2a; November 2019−Kruskal−Wallis chi-squared = 2393.1, df = 6, p-value < 0.001; March 2020−Kruskal−Wallis chi-squared = 804.64, df = 6, p-value < 0.001, see supplementary materials for full details) and prevalence (>78%; Fig. 2b; November 2019−Kruskal−Wallis chi-squared = 2422.5, df = 6, p-value < 0.001. March 2020−Kruskal−Wallis chi-squared = 883.55, df = 6, p-value < 2.2e−1, see

supplementary materials for full details) at both sites. The bleaching prevalence and severity values recorded on MCEs here are comparable to those measured globally on shallow-water reefs during several previous mass bleaching events[21–23]. During March 2020, five months later, coral bleaching at MA at mid- and lower-mesophotic depths was significantly less severe and widespread compared to November 2019, indicating partial overall coral community recovery (Fig. 2a, b).

The shallow-water reefs in the Chagos Archipelago experienced high mortality following the 2014−2017 global-scale bleaching event, with coral cover decreasing to <10% on many reefs[24,25]. The highest mortality was recorded on reefs at depths <15 m (our surveys started at 15 m deep), and coral cover on shallow-water reefs at Egmont Atoll was observed to increase between 2012 and 2016[24,25]. Further recovery of the reefs in the Archipelago has been observed in subsequent years[26]. Despite the high mortality experienced on reefs following the 2014−2017 bleaching event, over 3000 live colonies from 36 genera and 127 morphospecies of scleractinian corals were surveyed between 15 and 20 m in the current study (Supplementary Table S1). Thus, the low prevalence and severity of bleaching observed on reefs between 15 and 20 m in the current study is unlikely to be due to the low number of colonies present at this depth.

Shallow-water reefs in the Chagos Archipelago may now be composed of a higher proportion of heat-tolerant genera and species of scleractinian corals than before the mass bleaching events of 2014−2017, which is possibly reflected by the low prevalence of bleaching recorded on shallow-water reefs in the current study.

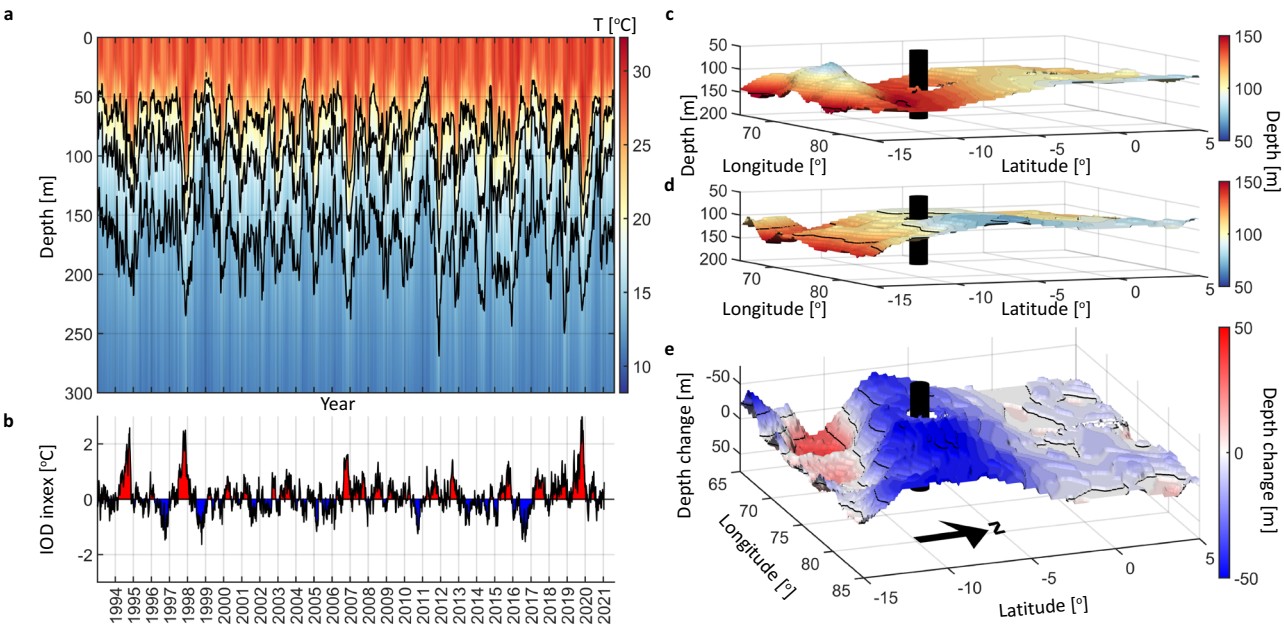

**Fig. 3 | Central Indian Ocean temperature evolution. a** Temperature throughout the upper 300 m within the central IO between 1993 and 2021 derived from the Copernicus Marine Services Global Reanalysis numerical model and, **b** the corresponding IOD index; **c** model-derived depth of the 22 °C isotherm, indicative of the thermocline depth, during November 2019 in the Chagos Archipelago (black cylinder) and **d** March 2020, and **e** the difference in thermocline depth between the two periods.

However, the thermal stress of surface waters was not detected in the four months preceding and the four months subsequent to our cruise by the modelled local temperature fluctuations in the Archipelago between 2016 and 2021 (supplementary Fig. S3), in addition to NOAA Coral Reef Watch[18] (although this metric must be taken with caution[19]. Previous studies within the Archipelago have shown that temperatures above 29.5 °C can induce bleaching[24]. During our surveys in 2019, the CTD mounted on the ROV continuously monitored temperature throughout the dives and recorded temperatures above 29.5 °C (in water <25 m) in only two of the 12 dives conducted during our expedition (supplementary Fig. S2; Dive 14 at IDR and Dive 18 at MA). The water temperature during all other dives remained below 29.5 °C for the duration of the dives. Thus, the lack of bleaching observed on shallow-water reefs (15–20 m) is likely due to an absence of thermal stress rather than a lack of coral colonies on the reef and/or a high proportion of thermally tolerant individuals. As mesophotic corals live in low irradiance zones, and could potentially be affected by a change in water transparency, irradiance data with depth collected in 2020 and 2022 are also provided to the reader for their perusal in Supplementary Table S7, while a difference of >10% in cloud cover has been observed between the two sampling periods.

On shallow-water reefs, high bleaching variability at local scales has been found to be partly due to differences in species composition in terms of both taxa and abundances[22,27–29]. In this study, significant differences in coral community composition were observed between depth bands within each site (PERMANOVA test on abundance data, 999 permutations, results in Supplementary Tables S2-1, S2-2). However, less difference in community structure was observed between the two sites at the same depth compared to two different depths within a site (Supplementary Tables S2-1, S2-2), and clear groupings of communities from the two sites at the same depth are shown in Fig. S1. At the same depth, the two survey sites share a high number of coral species (Table S3) and it is likely that the significant difference in coral community structure detected in the PERMANOVA is driven by the presence of a low number of rare coral species. Furthermore, the few coral species that span the depth distribution from upper-mesophotic

waters (30–40 m) down to the lower mesophotic zone (60–70, and 80–90 m for *Leptoseris* spp.) experienced more severe bleaching in terms of prevalence and severity at mesophotic depths, with more coral bleaching observed at MA compared to IDR (Supplementary Table S4), indicating that coral composition is not the only driver of differences in coral bleaching along the depth gradient.

## Thermocline deepening

The unprecedented depth to which coral bleaching occurred at Egmont Atoll during 2019 was likely attributable to the deepening of the thermocline that led to the presence of warm surface waters at a depth normally associated with the thermocline. This subsurface internal response is entirely distinct from the usual indicator of coral bleaching threat, which is essentially a function of SST alone. Analysis of the NOAA CRW database[18] for the Chagos Archipelago region and wider central Indian Ocean over the four months before and after November 2019 confirmed that no coral bleaching warning was recorded within this period, in addition to the modelled local temperature fluctuations in the Archipelago between 2016 and 2021 (Supplementary Fig. S3). Thus, it would normally be assumed that there was no thermal stress (or associated bleaching) in shallow-waters (<30 m) around Egmont, demonstrating that SST is likely a poor predictor of coral bleaching at mesophotic depths[30].

Instead, the observed bleaching during 2019 coincided with the strongest positive phase of the IOD observed since records began (Fig. 3b), causing an enhancement in easterly winds throughout the eastern and central Indian Ocean and a sustained westward current throughout the surface mixed layer. The central and western Indian Ocean thermocline deepened[31] (Fig. 3a), rendering coral communities that usually reside within the thermocline susceptible to bleaching through exposure to warmer surface waters[32]. Specifically, the 22 °C isotherm deepened from a depth of 60–140 m throughout a period of 3 months centred on November 2019. The depressed thermocline subsequently propagated westwards towards critical coral habitats in the Seychelles and Mauritius, where similar impacts on deep reefs would be expected. Five months later, as the IOD transitioned to a

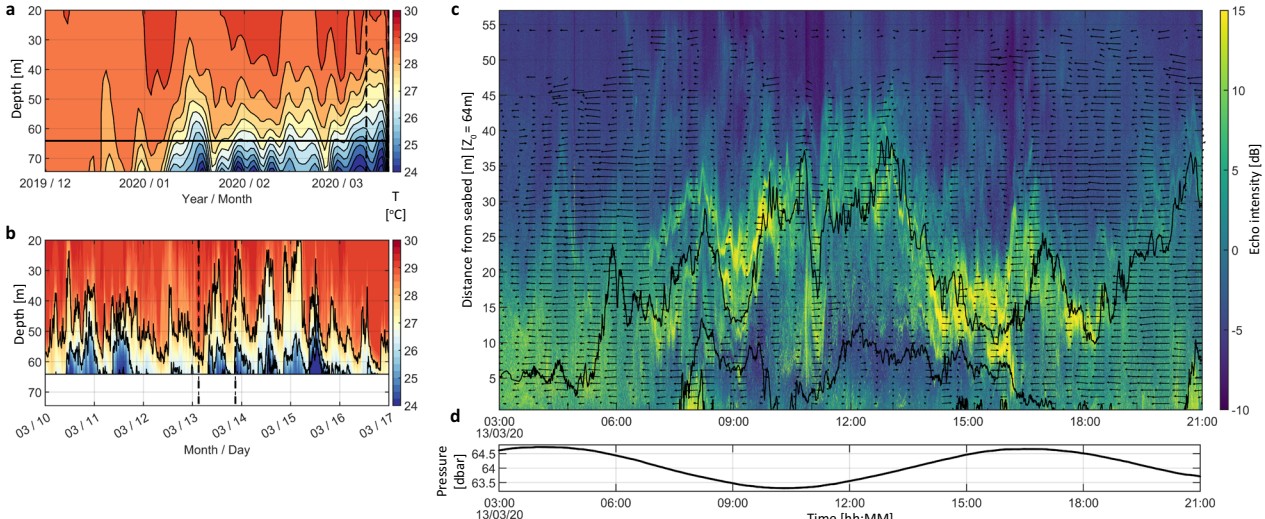

**Fig. 4 | Local temperature fluctuation at Egmont Atoll. a** Temperature evolution with depth over the period December 2019–March 2020, indicating the shoaling of the thermocline as the IOD changed phase from strongly positive to weakly negative, **b** the same but displayed at the higher resolution for a one-week period during March 2020 to illustrate the substantial vertical excursions of the thermocline at tidal frequencies and **c** echo amplitude (6 mm vertical resolution) and **d** pressure (indicating tidal height) measured by the ADCP deployed in Manta Alley in a water depth, $z_0$, of 64 m over an 18 h period. Note the high echo amplitude associated with the evolution of the thermocline in response to the tidal forcing, for which the prevailing currents are indicated by the black vectors; high echo amplitude suggests the presence of turbulence and/or biological material advected by the waves. Also included are the 22 and 23 °C isotherms to demonstrate the elevated turbulence, apparent as high echo amplitude, along the most steeply sloping isotherms where shear instability is enhanced.

weakly negative phase, the thermocline had recovered to more typical depths of 40–50 m and the coral at depth displayed less evidence of bleaching (Fig. 3c–e).

## Local-scale variation in bleaching influenced by internal waves

The differences in bleaching severity and extent between MA and IDR suggest that, whilst the basin-scale impact attributable to the IOD deepened the thermocline throughout the region, a local influence further modulated the thermocline depth and relieved thermal stress at IDR. To investigate the source of this variability, we analysed data from sub-surface moorings equipped with oceanographic sensors to measure current velocity and temperature throughout the entire water column at both sites. The observations were complemented by and contributed to the validation and tuning of, a high-resolution numerical model that further benefitted from a multibeam bathymetry survey of the entire atoll to depths of 400 m that we conducted prior to the ROV surveys in November 2019.

The thermocline in MA shoaled as expected between November 2019 and March 2020 (Fig. 4a) but, when observed over shorter periods, also underwent vertical excursions of >20 m amplitude at tidal timescales (Fig. 4b). As opposed to the frequently observed generation of internal tides by cross-slope orientated currents over continental slope margins, the internal waves at MA were coincident with signals in along-slope currents; the resulting near-bed temperature fronts varied in their character, resembling at times turbulent bores and, at other times, nonlinear wave trains. Furthermore, increases in near bed temperature (evident as downward deflected isotherms in Fig. 4c, corresponding tidal height in Fig. 4d) are accompanied at shallower depth by decreases in interior temperature, with isotherms deflected upwards. Such behaviour is consistent with Mode 2 internal waves, a less frequent observation throughout the ocean where attention frequently highlights the importance of Mode 1 waves within which the direction of vertical movement of isotherms is consistent throughout the whole water column.

The numerical model output, supported by the moored observations, confirms that the headland at IDR is characterised by a notably different internal wave regime, with waves of larger amplitude which tend to draw cold water up the slope rather than depress warm water downwards (Supplementary Fig. S8). Thus, in a time-averaged sense, the internal waves could enhance warming at the depth of the thermocline in MA where bleaching was observed by preferentially advecting warm surface waters downslope. At IDR, the waves could have an opposite effect by predominantly advecting cold, sub-thermocline water up the slope and relieving thermal stress in the benthic environment.

To understand the discrepancy in internal wave activity and coral bleaching between the two sites, the numerical model was forced with tides at both diurnal and semi-diurnal frequencies following analysis of the long-term current data (see modelling methods). The model was run for a period of 15 days, to include a spring-neap cycle, initialised with a vertical temperature profile obtained in the deep water surrounding Egmont Island that represents oceanographic conditions undisturbed by internal waves over the slopes. The model replicated the observed pattern of temperature variability within the thermocline in terms of the frequency and amplitude of perturbations (Fig. 5c, based on in-situ observations, Fig. 5d). Most significantly, the Mode 2 internal wave activity causes a sustained increase in the time-averaged temperature at the seabed relative to the undisturbed regime within MA at the depths where bleaching was most pronounced, between 60 and 90 m deep (Fig. 5a; Supplementary Fig. S5). The same impact was absent at IDR due to the cooling, rather than warming, influence of internal waves on the northern flank of the atoll, demonstrating the highly localised potential influence of internal waves on coral bleaching within MCEs (Supplementary Figs. S4, S6, S7).

The temperature measured by the ROV during the dives to survey MCEs corroborates this result. In-situ temperatures at the seabed were consistently higher than the background values obtained from a vertical profile obtained nearby but in deeper water away from the influence of the slope (Fig. 5b, supplementary Fig. S5b; not as pronounced for IDR, Fig. S6b). Thus, internal waves could elevate the time-mean temperature at a depth beyond the values that would be experienced by corals without any tidal forcing, exacerbating the impact of basin-scale dynamics and potentially modulating bleaching in MCEs over horizontal distances of $O$ (1 km). Additional impacts may

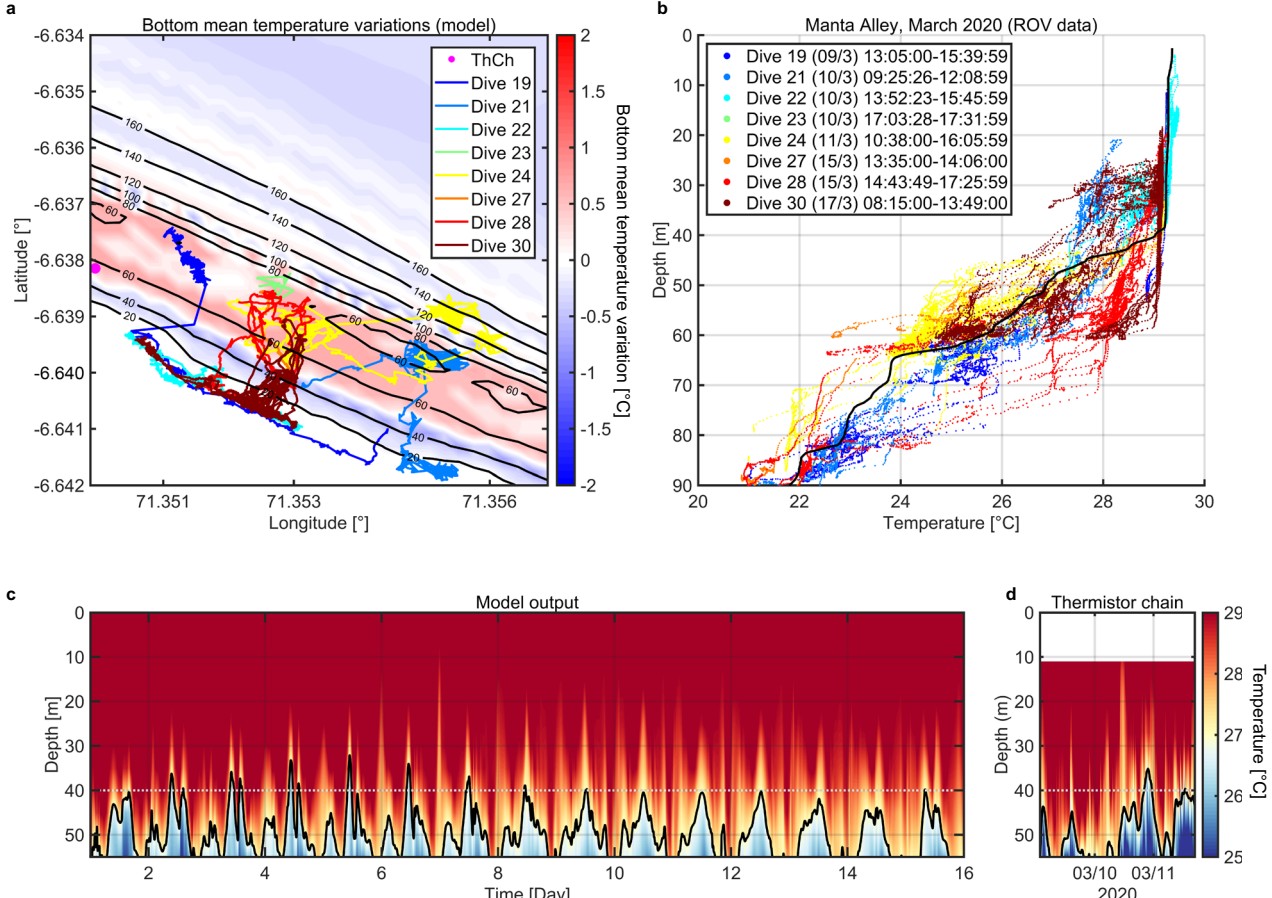

**Fig. 5 | High-resolution numerical modelling at Manta Alley in March 2020.**
**a** Tracks of the ROV dives in Manta Alley in March 2020. The colours of the lines show the ROV dives marked in the legend. The isobaths (solid lines with the labels in meters) are overlayed with the model-predicted bottom temperature (°C) variations calculated over a 15-day cycle. **b** Water temperature (°C) throughout the depth gradient in Manta Alley, March 2020. Coloured lines represent ROV in-situ data and coincide with panel A. The positions of the ROV transects are shown in (**a**). The solid black line shows the background temperature (°C) profile recorded off Egmont Atoll in March 2020. **c** Model predicted temperature time series in the position of the mooring (shown in panel **a** by a magenta dot). **d** Temperature (°C) time series recorded by the thermistor chain at the position of the mooring.

also be generated by remotely forced internal waves that, after generation at other sites, radiate out into the surrounding ocean before, potentially, breaking over the slopes surrounding islands such as Egmont.

In summary, our results demonstrate that scleractinian corals at mesophotic depths from the Chagos Archipelago, in the Indian Ocean, are susceptible to thermal stress and bleaching in the same manner as that experienced by shallow-water reefs. This is in contrast to previous studies on mesophotic corals from different geographic areas[27,33–35], highlighting the need to incorporate these unique ecosystems within conservation planning[13,36,37]. Our study adds further weight to the increasing evidence demonstrating that MCEs are not a universal refugia for shallow-water coral reefs[12,37–40]. In addition, SST as a measure of thermal stress is a poor indicator of coral bleaching at mesophotic depths in the Indian Ocean, revealing the urgent need to consider the impact of subsurface dynamical processes on thermocline depth[41]. Here, the IOD was potentially the dominant mechanism responsible for coral bleaching at a regional, monthly, scale and is projected to increase in frequency and severity with global warming[42], similarly to ENSO in the Pacific region. At local scales, internal waves, especially Mode 2, were suggested to have an impact on the thermal regime at mesophotic depths and could be responsible for deep bleaching over distances <1 km, which realistically requires high-resolution numerical modelling to fully understand. As shown in this paper, the internal wave influence on coral bleaching is sensitive to the details of the bathymetry, stratification and forcing, all of which vary over both space and time. At different locations, other modes of variability may play a significant role in modulating the near-bed temperature regime. Currently, remotely sensed products and/or numerical output with typical resolutions of 5 km are used to evaluate the threat to surface corals from bleaching and are considered state-of-the-art, yet are clearly deficient for resolving subsurface dynamical processes at local, atoll-scales. Concernedly, it is likely that many mesophotic bleaching events go unnoticed due to the lack of appropriate monitoring techniques for deeper reefs. As researchers spend increasing amounts of time on MCEs, records of bleaching events at mesophotic depths are likely to increase. However, numerical models that are able to incorporate dynamical processes at local scales will further our understanding of the frequency and severity of bleaching on MCEs. To conclude, additional data on the composition, dynamics and threats to MCEs are required to better understand these critical ecosystems and to develop sound conservation and resource management criteria.

## Methods
### Study area
This study focuses on Egmont Atoll, within the Chagos Archipelago (Fig. 1), where two research cruises were undertaken in November 2019 and March 2020. Egmont Atoll has a total surface area of 40 km² and is composed of six islands and several distinct areas[43]. Mesophotic coral ecosystems were investigated at two sites: Ile Des Rats (IDR) on the

North-western coast, and Manta Alley (MA) (also called Egmont Mid) along the North-eastern coast (Fig. 1).

## Biological data

**Data collection.** A Falcon Seaeye Remotely Operated Vehicle (ROV) was used to collect underwater imagery at all sites. The ROV was equipped with four SAAB Seaeye LED daylight white lamps (34 W, 3520 Lumens), with two mounted on a tilt platform with a SAAB Seaeye camera in the centre (recording live video in standard resolution (720p) with wide-angled lens (91° in water), functioning with low light, with an information overlay displaying depth, time, heading, pitch and roll). A second camera (GoPro Hero 4) was positioned directly below the Seaeye camera and recorded footage in High Definition (2.7k, 24 fps, wide field of view). Dive duration was limited by the GoPro battery length, which was ~3 h. In addition, a Valeport Modus conductivity–temperature–depth (CTD; serial number 31027) was mounted on the ROV (calibrated on 23/04/2015 and 10/12/2020 with no modification required between the two dates).

Due to the presence of a strong current at many sites in the Chagos Archipelago, it was difficult to maintain a constant heading at an altitude close to the seabed to undertake linear video transects. Thus, still image samples were collected along transects. To collect image samples, the ROV descended to the appropriate depth and approached the seabed (altitude <1.5 m) with the cameras positioned at an oblique angle to the seabed. Still, frames were captured in order to collect high-quality images that allow specimen identification and maximise the field of view of the seabed. While digital imagery has its limitations, non-destructive image collection techniques such as those used in this study are regularly employed to quantify benthic habitat composition in shallow and deep-sea ecosystems and contribute significantly to surveying and monitoring vulnerable habitats[44–47].

For each image, the time, depth, latitude and longitude of the sampling position were recorded, and 90 images were collected per depth, with a total of 1080 images from the two sites. Along the depth gradient, six depth bands, chosen a priori and based on the literature, were surveyed to cover the shallow and mesophotic zone: 15–20; 30–40; 60–70, to 80–90, 110–120 and 150–160 m. Depth bands were also chosen to facilitate ROV operation and counteract the strong upwelling and downwelling currents occurring in the Archipelago. The two deepest depth bands were not included in the analysis for this study as scleractinian corals were not observed at these depths. Surveys were undertaken along the entire depth gradient at both IDR and MA in November 2019. In March 2020, only surveys at MA were possible due to an extremely dynamic current regime close to the atoll, directed from West to East, preventing safe access to IDR. Scale, enabling the calculation of image area, could not be quantified due to technical issues related to the ROV lasers. However, the collection of benthic imagery was standardised by ensuring that the camera angle was consistent throughout the dives and ensuring the ROV was as close to the seabed as possible when still images were captured. Hence, this study focuses on the proportion of bleached colonies observed from the total number of colonies surveyed at each depth, rather than an area-based survey.

Coral bleaching prevalence (number of individuals in the population affected, %) and severity (bleaching extent within a single colony, bleaching index, BI) were assessed in November 2019 and March 2020, using still images extracted from video transects, from shallow and upper mesophotic (15–20 and 30–40 m) to mid and lower mesophotic depths (60–70 and 80–90 m).

**Image analysis.** Still images collected along each transect were annotated using the online annotation platform, BIIGLE[48]. All benthic organisms >1 cm within each image were identified and quantified (Fig. 6). All organisms identified as distinct morphospecies were assigned an

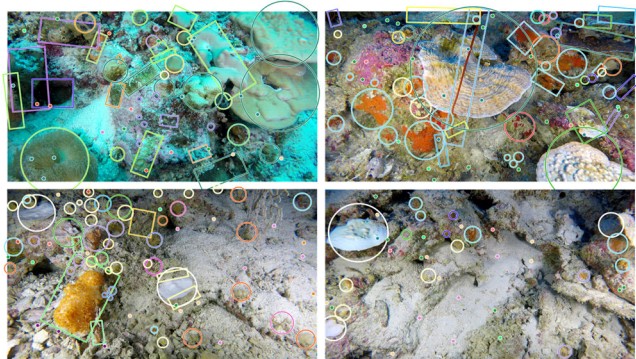

**Fig. 6 | Annotated images.** Examples of images collected using the ROV after being annotated in Biigle software. Ile Des Rats (IDR), 15–20 m deep (top left); Manta Alley (MA), 60–70 m (top right); IDR, 80–90 m (bottom left); MA, 80–90 m (bottom right). Different shape colours indicate different morphospecies.

Operational Taxonomic Unit (OTU) and were identified to the highest taxonomic resolution possible. However, morphospecies may correspond to species, genus, family or higher taxonomic levels. Alongside the image analysis, a morphospecies catalogue was created for the region following the global standardised marine taxon reference image database suggested by Howell et al.[49]. For the purposes of this study, only morphospecies identified as zooxanthellate corals were analysed.

## Hard coral bleaching identification

**Bleaching extent and severity.** Edmunds et al.[50] indicated that subjective rankings of bleaching based on coral colour are strongly associated with *Symbiodinium* density, chlorophyll *a* concentration, and the intensity of green, blue and red colour spectrums. Subjective ranking may therefore be a good indicator of disruption of the symbiosis for bleaching events[23,50]. Thus, the bleaching extent within a single colony, also called bleaching severity, can be calculated based on coral colour. In addition, a bleaching index (BI) can be used to assess a species' overall health. Following previously published methods[23,50–52] 4 bleaching categories were created in order to classify each scleractinian colony observed in the images to provide a precise measure of coral health: (1) unaffected (normal coloration), (2) light (combining pale, a lighter colour than usual for the time of year, and 0–20% of the coral surface bleached), (3) moderate (20–80% bleached) (4) severe (80–100% bleached and bleached with partial mortality). Total mortality was not taken into account to avoid counting corals that had not bleached during this event. The four categories are identical to those used in a previous mesophotic study[51]. As the images did not include a uniform Lambertian reference material, a combination of methods was used to assign colonies to these four categories: the still image itself when fully saturated by light; the video footage from which the images were sampled to verify the bleaching category of a colony when the image itself was not sufficient; different bleaching severities within the same image, thus exposure to the same light conditions; physical samples of hard coral from different species and different depths with the corresponding underwater image; and both images and physical samples of healthy corals collected in a subsequent expedition in 2022, to compare with the samples of the present study.

The bleaching index (BI) was calculated from the percentage of coral colonies observed in each of the above four bleaching categories:

$$BI = (0_{C1} + 1_{C2} + 2_{C3} + 3_{C4})/3 \qquad (1)$$

where $_{Ci}$ is the percentage of observations in each of the above four bleaching categories.

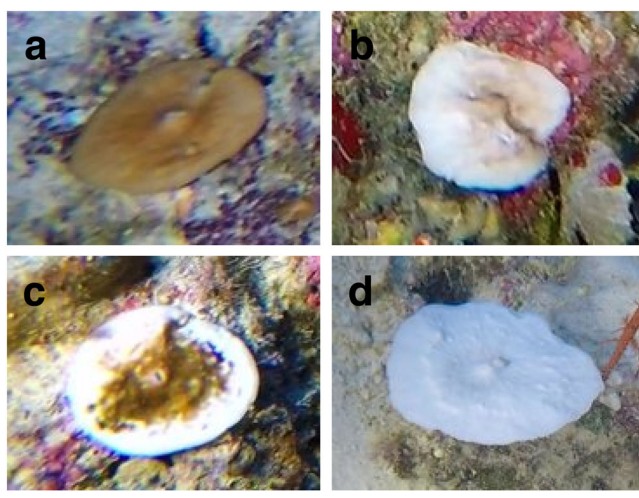

**Fig. 7 | Bleaching categories using examples of *Leptoseris* sp. a** Unaffected,
**b** light, **c** moderate, and **d** severe.

These categories were created in BIIGLE software and assigned to every zooxanthellate scleractinian coral identified during the image annotation (Fig. 7).

**Bleaching prevalence.** Bleaching prevalence has been used by several authors in coral bleaching studies, and is the number of individuals in the population affected (see references in Quimpo et al.[28]). Here, bleaching prevalence was calculated as a percentage by dividing the number of bleached colonies by the total number of colonies recorded and multiplying by 100, using the same categories in BIIGLE described above.

## Statistics
In order to detect differences in terms of coral bleaching between the two sites, the two seasons and the six depth bands, using R software[53], a Kruskal–Wallis test was performed with pairwise comparisons using post-hoc Wilcoxon rank sum test with continuity correction and Benjamini–Hochberg (BH) adjusted method for both BI and bleaching prevalence calculations. Differences in hard coral community composition between sites and depths were investigated using a multi-dimensional scaling (MDS) plot and permutational analysis of variance (PERMANOVA) test on abundance data, square-root transformed and Bray–Curtis similarity, with a dummy variable added to account for zero-inflated data[54–56] using PRIMER v6. A test was significant when the $p$-value was <0.05.

## Sea surface temperature via NOAA satellites
Degree heating weeks (DHW) data were extracted from the National Oceanic and Atmospheric Administration (NOAA) Coral Reef Watch (CRW) database[18] for the Chagos Archipelago region (−8 to −4°S; 70 to 73°E) and wider central Indian Ocean (−30 to 20°S; 40 to 100°E). DHW shows accumulated heat stress (HotSpots) by accumulating the instantaneous bleaching heat stress during the most recent 12-week period, with significant coral bleaching usually occurring when the DHW value reaches 4 °C-weeks[18]. However, this metric should be taken with caution as research has shown that its ability to predict bleaching varies with geographical location[19].

## Moored measurements
Data were obtained at two locations within Egmont from three separate moorings, two collocated in MA, and one at IDR. The current meter mooring within MA consisted of a near-bed ($z = 2$ m above bed) upwards looking Signature 500 kHz ADCP, whilst a secondary mooring located within 100 m horizontally comprised a taut-line mooring on which were mounted two RBR Concerto conductivity–temperature–depth (CTD) sensors at the top and bottom of the mooring and 23 RBR solo-T temperature sensors spaced at 2 m vertical intervals between $z = 4$–50 m. At IDR a single taut-line mooring contained temperature-depth sensors (top and bottom), and 25 temperature sensors spaced at vertical intervals of 2–4 m, and an in line upwards looking Aquadopp 400 kHz ADCP. All thermistors and temperature–depth sensors were logged at 1 Hz, and CTDs logged at 0.2 Hz. The Nortek Signature ADCP operated on a dual sampling regime, providing 10 min average velocities at 2 m vertical intervals and 23 min of 1 Hz burst velocity data in 1 m vertical bins and 1 Hz echosounder data at 6 mm intervals at the start of every hour. Temperature data were interpolated to regular 2 m vertical intervals and cleaned in both spatial and temporal dimensions with a maximum deviation filter and running mean window. Noise issues in the echosounder data necessitated correcting with a dynamic level adjustment to a common noise floor as well as gain normalisation via a logarithmic fit for acoustic decay with range. In the case of Fig. 4a temperature data have been binned to 1-day samples and further low passed with a 6-day cut-off to better visualise long-term evolution.

## Remote sensed data
Indian Ocean dipole (IOD) index data is taken from the Ocean Observations Panel for Climate Dipole Mode Index data set (https://psl.noaa.gov/gcos_wgsp/Timeseries/DMI/). Remotely sensed temperature is taken from the Copernicus Marine Service Global Ocean Forecast (from 2019) and Reanalysis (before 2019). Time series were extracted from 8.125°S and 73.875°E in deep water to the east of the Archipelago to minimise the effect of perturbations arising from flow–topography interaction in the Chagos Archipelago and highlight the basin-scale evolution of the thermocline due to the IOD.

## High-resolution oceanographic numerical model
Modelling the tidal motions in Egmont Atoll was conducted using the nonlinear nonhydrostatic Massachusetts Institute of Technology general circulation model (MITgcm)[57]. The topography used in the model is presented in Fig. 8A; bathymetric data were collected during a multibeam survey of Egmont to depths of 400 m during 2019 and gridded to 5 m horizontal resolution. Beyond depths of 400 m, data were interpolated to match the GEBCO bathymetry. The vertical model resolution was 5 m and the horizontal resolution in the central part of the model domain was 25 m in both horizontal directions. The model simulation encompassed 16 days to allow for a day of spin-up and 15 days of predicted currents and temperature variability. Reflection of barotropic tidal waves from the domain boundaries was excluded through the application of Orlanski-type boundary conditions.

According to the TPXO inverse tidal model predictions[58] and in-situ data from ADCP moorings the tidal currents in the area consist of comparable values of diurnal and semi-diurnal harmonics. Figure 8B and C show depth-integrated zonal ($u$) and meridional ($v$) velocities without long-term oscillations at MA. Both currents show a 15-day periodicity that results from a combination of semidiurnal $M_2$ and diurnal $K_1$ tidal harmonics.

The combination of semidiurnal $M_2$ and diurnal $K_1$ tidal harmonics was activated in the model using a methodology described in Vlasenko and Stashchuk[59]. The temperature/salinity profiles were taken from CTD profiles acquired in deep water, away from the internal wave activity over the slopes, during 2019 and 2020 near Egmont. The observed vertical profiles represent the undisturbed fluid stratification required for the model initialisation. The temperature profile for March 2020 is shown in Fig. 5b by the black solid line. The modelled temperature, salinity, and velocities were compared with the ADCP mooring and thermistor chain data in MA. Figure 8D and E show depth-averaged time series of $u$ and $v$ velocities at the same position as the

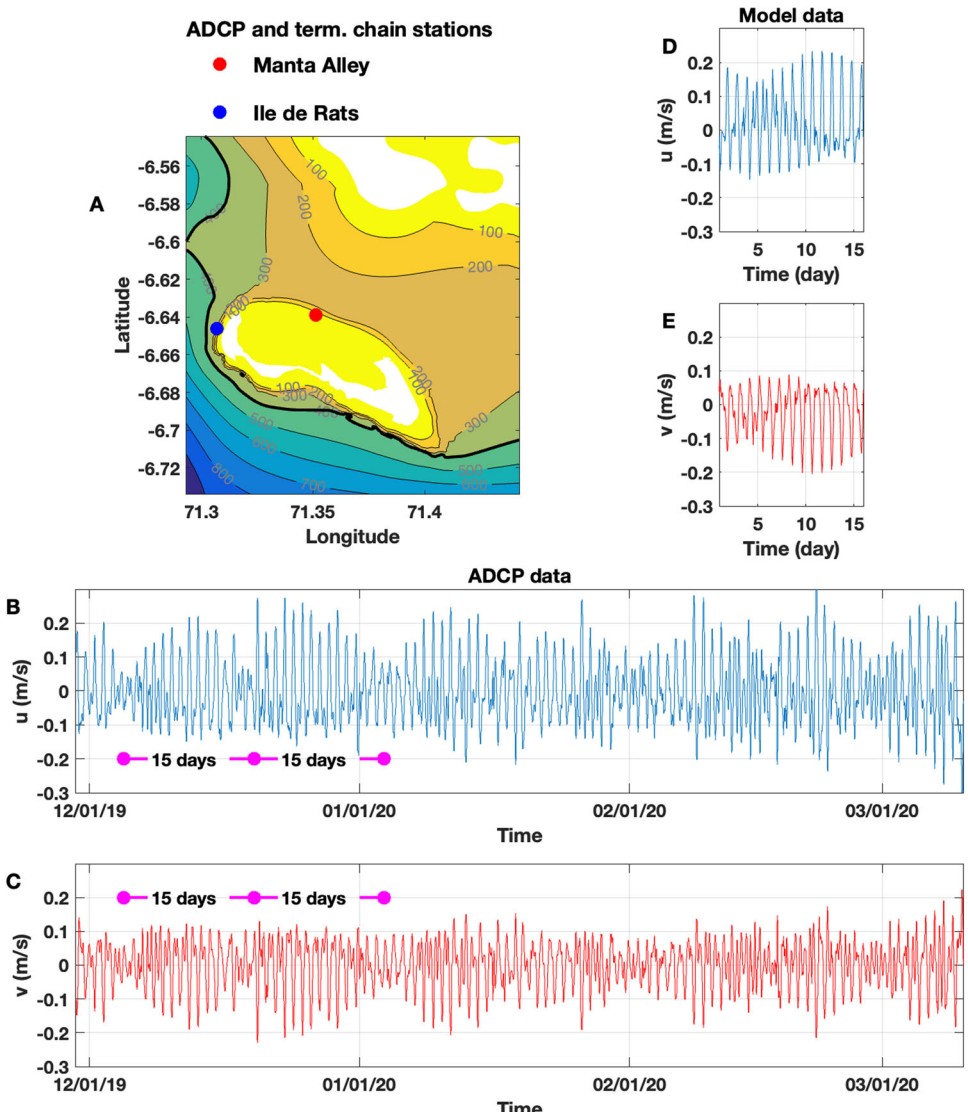

**Fig. 8 | Comparison of tidal in-situ and modelled data. A** Bottom topography in Egmont Atoll with the position of the deployed mooring (red dot). **B** and **C** Time series of zonal ($u$) (**B**) and meridional ($v$) (**C**) tidal velocities recorded at the mooring. The background non-tidal currents were deleted from the original raw ADCP records. **D** and **E** Tidal currents reproduced by the model at the position of the mooring.

ADCP, Fig. 8B and C. From the coincidence between Fig. 8C, B and D, E, it is seen that the model correctly reproduces tidal currents in terms of phase and amplitude.

Depending on the combination of buoyancy frequency $N(z)$, bottom topography $H(x)$, Coriolis parameter $f = 2\Omega \sin \theta$ ($\Omega$ is the angular velocity of the Earth and $\theta$ is the latitude), and the tidal velocity $U_m$, a variety of generation scenarios are possible[60]. The periodic response in time, $t$, with frequency, $\sigma$, of internal waves propagating horizontally in the ocean along the $x$-axis with wave number $k$ are given as $q(z) = \sin(kx - \sigma t)$. In terms of the boundary value problem, the vertical structure $q(z)$ of these waves is defined by the following boundary value problem (BVP):

$$q_{zz} + k^2[(N(z)^2 - \sigma^2)/(\sigma^2 - f^2)]\,q = 0 \qquad (2)$$

$$q = 0; \text{ at } z = 0; z = -H \qquad (3)$$

Note that the above-mentioned BVP has solutions, $q_1(z)$, $q_2(z)$, $q_3(z)$, … with horizontal wave numbers $k_1$, $k_2$, $k_3$, …, that define the vertical structure $q_i(z)$ and horizontal scale $k_i$ ($i = 1,2,3,…$) of generated internal waves. The BVP shown above is applied to a very general case which includes both strong and weak Earth rotation. For the Chagos Archipelago, the Coriolis parameter $f$ at the latitude of Egmont Atoll, 6.5º, is small compared to the tidal frequency $\sigma$. If so, the direct theoretical consequence from the BVP for the considered area is that the properties of internal waves are the same for both semidiurnal and diurnal baroclinic tidal constituents.

### Reporting summary

Further information on research design is available in the Nature Portfolio Reporting Summary linked to this article.

### Data availability

The datasets supporting this article have been uploaded as part of the supplementary materials. The ROV data used for the modelling are available here: https://figshare.com/articles/dataset/Egmont2020/19925381. Oceanographic data not provided are being analysed and further developed by current project students but are available upon request from P.H. (phil.hosegood@plymouth.ac.uk). All requests will be responded to within one week of receipt. Source data are provided with this paper.

## Code availability

Codes are not provided as they are being analysed and further developed by current project students. They are however available upon request from P.H., phil.hosegood@plymouth.ac.uk; C.D., clara.diaz@plymouth.ac.uk; or N.L.F., nicola.foster@plymouth.ac.uk. All requests will be responded to within one week of receipt.

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

## Acknowledgements
We thank the United Kingdom Foreign and Commonwealth Office and the British Indian Territory Administration for allowing us to undertake this research and the Garfield Weston Foundation and the Bertarelli Foundation for funding this work and supporting data collection in the Chagos Archipelago in the Indian Ocean. We also thank Nataliya Stashchuk and Vasyl Vlasenko for providing the model output and preparing the modelling figures. We finally thank the captain, Craig Henn, and the crew of the Tethys Supporter for their assistance throughout the cruises.

## Author contributions
N.L.F., C.D., P.H., and K.L.H. conceived the study. C.D., N.L.F., P.H., K.L.H., E.R., P.G. and A.B. conducted the fieldwork and collected the data. C.D. analysed the biological data with support from N.L.F., P.H. and E.R. analysed the oceanographic and modelling data. A.B. processed the multibeam data. C.D. led the manuscript, with substantive contributions from P.H. and N.L.F. M.J.A. and all other authors contributed data and made significant contributions to the text.

## Competing interests
The authors declare no competing interests.
