## [Peer Review File · Nature Communications]

REVIEWER COMMENTS

Reviewer #1 (Remarks to the Author):

Review of Deep Coral Bleaching Driven by Hidden Changes in Thermocline Depth for Nature Communications

I am now reviewing this revised manuscript for the second time and as previously stated I am impressed with the physical oceanography and its causal role in the observed bleaching of scleractinian corals on MCEs in the Indian Ocean. I also appreciate the authors response to the previous reviewer's comments. Nonetheless, I still believe there are issues that should be addressed.

Line 18; Suggest this reads as follows; "Our results demonstrate the potential vulnerability..."

Line 26; Here, and throughout the manuscript "Scleractinian" is not capitalized. It is not a taxon and should be in lower case.

Line 27; Suggest this reads as follows; "...sea surface temperatures (SSTs) and subsequent coral bleaching; current estimates..."

Line 32; Suggest this reads as follows; "...depths of 30-150 m and have been estimated to occupy..."

Line 33-34; Suggest this reads as follows; "It has been hypothesized that these communities are buffered against anthropogenic changes, and therefore might represent a refuge that replenishes shallow-water coral species with vertically dispersed larvae^{8,10}. Or words to that effect.....

Line 51; "in-situ" is not hyphenated and should be italicized as it is Latin.

Line 53; Per my previous review. The fact that a flawed metric (i.e., DHW) is being used by a large proportion of the community is not a justification for its continued use. If the authors insist on using this metric, knowing these flaws as I pointed out in my last review, then they must at least tell the community that there are known problems, with references, especially as it relates to its predictive capabilities.

Line 63; Reviewer #2 had some keen insights on this, and I will be interested in seeing how the authors deal with the issue of post-bleaching coral communities being dominated by “winners” as a reason for little to no bleaching on shallow reefs compared to potentially more thermally sensitive corals at MCE depths. The recovery you discussed in response to the reviewer, in the absence of any seascape genetic data, was likely a result of recruitment from local populations, or the winners so likely more resistant to thermal stress. Also, 3000 corals in 2019 and 1000 in 2020 on shallow reefs-what happened here?

Figure 1. The euphotic zone, classically defined in the oceanographic community is 0-200 m and the mesopelagic zone is 200-1000 m. The MCE depth definition is placed, or should be placed, over these long accepted oceanographic depth descriptions as near coastal benthic habitat.

Line 110; “5” should be “five”

Line 128-135; There it is-the “winners” argument, but then you try and support your interpretation with a flawed metric. Again, if you are going to do this you must state clearly for the readers the flawed aspects of the metric.

Figure 2; Need a fuller description of statistics. As requested in my previous review what primary analysis, and post hoc analysis, was run? What were the DF, F-value, p value corrections for multiple comparisons, etc. as applicable.

Lines 155 and elsewhere; ANOSIM provides a test to detect differences in two or more groups of “sampling units” based on ranked similarities. First, why not present the results as an NMDS or cluster plot? Would be much easier for the reader to see the depth comparisons of coral communities between depths and sites. Did the authors ensure that the heterogeneity of dispersion, which ANOSIM is very sensitive to, is equivalent or at least similar between depths and sites? Also, why the more permissive p value of 0.1 to start with? Is it possible that at a minimum p value of 0.05 some of these comparisons would not have been significant?

Line 165; “spp.” is not italicized.

Line 250; Is this supposed to say; "(See modelling methods)"?

Line 283; Suggest this sentence reads as follows; "In summary, our results show that scleractinian corals at mesophotic depths from the Chagos Archipelago in the Indian Ocean are susceptible to thermal stress, and bleach in a similar manner to that observed for shallow-water corals in contrast to other studies on mesophotic corals from different geographic areas²⁶, 32-34."

Line 286; Suggest this reads as follows; "Our study adds more evidence that MCEs are not, a priori, a universal source of coral recruits for replenishing shallow water coral reefs^{12, 13, 37-39}."

Lines 334-338; So, transects were employed. How long were they? How many images were taken at each depth? Were the same number of images taken at both sites at both times? What was the angle of the oblique imagery?

Lines 339-342; The statement that non-orthogonal, or oblique, imagery is taken by the deep-sea communities is true. But they usually collect those images, along with scale (!), and analyze them a priori knowing that different techniques (i.e., multiple lasers, Canadian grids) will be needed to obtain orthorectified images and accurate community characterizations for multiple ecological metrics such as abundance and percent cover. Regardless of whether plot or plotless designs are incorporated the imagery must be orthorectified.

Line 347; Why were the depth bands discontinuous?

Lines 362-372; You only present the "community" analysis on zooxanthellate corals for the presence/absence data. The additional information is unnecessary. Amend it for the analysis presented in the manuscript.

Line 386; "coral colour" is actually a function of its reflectance captured by the image under a specific light field. Without a reference, a control, based on a simultaneous image of a Lambertian surface you cannot accurately and consistently quantify image to image difference in "coral colour" even with a known light source. More on this later.

Lines 414-418; As stated previously; "Also, there is no indication of how bleaching prevalence was actually calculated because to compare between sites one would have to/should know the aerial extent

of the survey. That is the percentage of bleached coral at site A was 50% and site B was 75% based on a 100 m² survey. Or better yet do transects at each depth, even plotless transects would be fine as long as the areas surveyed were the same. Just like Quimpo et al. 2020, the paper the authors reference, did in their paper.

Response to authors comments of prior review

I appreciate the efforts of the authors to examine both irradiance and cloud cover at their sites as it relates to coral bleaching. I also understand that corals at mesophotic depths are probably naïve to temperature increases of 3-7°C. Figure C1 is helpful although changing the scale to allow a closer examination of the changes in irradiance at mesophotic depths would have been more informative. However, the authors do provide calculations of optical depths for the irradiance data they do have that are based on calculated K_d . The authors state that only 1% of surface PAR reaches 90 m so irradiance is not an issue as it relates to bleaching. But the corals at these depths are acclimatized to these extremely low irradiances. Any change in the optical properties may allow increased water transparency and increased irradiances, albeit small when thinking in terms of shallow coral reefs, but not inconsequential for extreme low-light acclimatized corals. For instance, the K_d for the 1% OD of 63 m is 0.073 while for 97.8 m the K_d is 0.047 from your table. These are not inconsequential differences and suggest significant changes in water transparency which could have consequences related to the role of light in bleaching at mesophotic depths, which on shallow reefs is primarily a light driven phenomenon; temperature sets the table, but light drives the process. The authors should give the reader the data and a chance to assess. Provide the OD data, give an example and say it is possible that these oceanographic processes, especially internal waves, could change the transparency of the water column and could have contributed to the bleaching process.

On the issue of using coral color as an index of bleaching extent and severity. As suggested above again, the calculation of a bleaching index based on coral color requires that all colonies being assessed in the image be orthogonal, and at the same height above the substratum so that the light source illuminates the coral evenly and at the same intensity. If this does not occur then this index, basically a measure of coral reflectance, requires, for each image, a reference to compare against. The reference material must be Lambertian in nature; a commonly used material is Spectralon, which can provide a diffuse reflection that is equal across all visible wavelengths. This allows the user to compare the reflectance of the standard to the reflectance of the subject under the same light source (and any differences associated with light intensity). For images that are not orthogonal such as used here multiple references would be needed since the distance from the light to the subject varies, and therefore so does light intensity across the image. This can be all entered in the authors BIIGLE program for appropriate annotation and is essential to compare coral bleaching between sites, and between years.

As far as bleaching prevalence I still believe that to compare between depths or sites one would have to/should know the aerial extent of the survey. That is the percentage of bleached coral at site A was

50% and site B was 75% based on a 100 m² survey. The authors describe taking images, and the number of images, but do not adequately describe the process-see comments above for Lines 334-338.

The authors contend that using the number of individuals to assess species diversity is a valid approach as described by Gotelli and Cowell 2001. Their techniques utilize rarefaction curves and I see no description of that in the M+M of this manuscript. Also, my reading of Gotelli and Colwell 2002 suggest that using species density, versus species diversity/richness would be a more appropriate metric to derive for quantitative studies, and therefore requires a knowledge of area analyzed. This is best accomplished using orthogonal images.

Again, the physical oceanography is amazing! The authors should couple that data to the bleaching prevalence data alone, once the M+M for the bleaching prevalence data is better explained, and omit everything else.

Reviewer #3 (Remarks to the Author):

See attached document

Review of NCOMMS-23-12279-T

I was brought in to review the physical oceanography reported in this manuscript. I have expertise in studying the dynamics of internal waves in the ocean using observations similar to those reported here.

In general, there is a reasonably detailed set of measurements of the physical state of the ocean at the two study sites. These observations come from moorings with temperature, salinity, and currents. Based on the description in the methods, the mooring design and the data that the moorings collected seems appropriate to assess the variability of temperature over the upper ~100 m with good vertical resolution on frequencies from minutes to weeks. Having this data is very good and an absolute prerequisite here. Unfortunately, however, the mooring time series start after (dec 2020) the primary bleaching period (nov 2020), and the data for whatever reason is given second billing to two numerical models.

MITgcm is used by the authors to model the internal wave variability. It is initialised by a CTD profile the authors collected, and forced by the semidiurnal and diurnal barotropic tides. The model is (sort of) compared to the data, and then the authors create a number of plots of model output that they use to argue about differences between the two reef sites. The authors settle on “mode 2” internal waves as differing between the two sites. I am fairly sceptical that the analysis of the model in the way presented here supports that conclusion to the exclusion of other hypotheses. I would have preferred to see more analysis of the data, and a deeper dive into the model insight about how the internal wave climate is different between the two sites.

It should be understood that my comments that follow are in an effort to get the best out of these data, and that I don't believe that the manuscript should be rejected due to the PO component of the paper. That said, I have a number of fairly significant concerns with the presentation and interpretation.

For the record, I actually do not believe that “definitively” identifying the presence of some internal wave pattern or another is a precondition for this being a valuable contribution. Simply identifying, in the data first, and the model second, that the different reefs have different internal wave climates is probably enough to POSE the hypothesis that this is the reason why one bleach and the other did not. More data/modelling would be needed to definitely show this.

Finally, there has been some fairly strident back-and-forth between the authors and the first two reviewers about the value of SST and “coarse” numerical models in predicting bleaching risk in mesophotic corals. My take is the following: of course a mode-derived “early warning” product tuned to the ocean surface mixed layer and shallow (<30 m depth) corals would have questionable application to 40-80 m depths. Is there actually debate about this? As the authors point out, because of the importance of internal wave variability, any global model driven early warning system is likely to be quite challenged. (As an aside, I will note that global models like mercator do not get the internal waves correct *anywhere* and so by definition can not at present capture this impact for any depth). Simply pointing out that current generation “bleaching warning systems” are not going to be as effective for mesophotic corals because of internal wave variability, without getting into the “difficulties” of

measuring the position of the thermocline – which is actually not that hard and is regularly done with moorings all over the world – is plenty of a conclusion enough in that realm for a paper like this.

I hope the authors find these comments useful. I have written many of them in what follows. It is not my position that every suggestion has to be followed up on. But I do worry that at present the PO story is incomplete and in some ways confusing and potentially even wrong. We want to get as much as that out of the paper as possible before it is published.

Major comments about the PO:

- 1) The presentation in Fig. 3 is confusing. First of all, Fig 3 is from a model reanalysis.. It is not data, it is the output of a model. The model does assimilate data, but it needs to be clearly understood that these are not actual observations, including within the figure/caption itself.

I have comments below about each subpanel.

- a) There really is no need to show the entire reanalysis period. Why is this done? I much prefer the presentation in Fig. S2. Fig 3 a and b are far too cluttered and this story is about 2019-2020, not the decade beforehand. I strongly suggest that this be re thought. It is almost impossible to understand anything about the temperature fluctuations in fig 3 a.
 - b) Fig 3 b should be zoomed into a much tighter time range. Again, there is little value in showing a long time series, since the point of the paper seems to be that 1) regional oceanography was anomalous (this has been shown by others and comes with a few references in the text) and 2) local effects matter. Neither fig 3 a or 3 b needs a decadal perspective.
 - c) Fig 3 c d and e are unnecessary. While somewhat interesting that the thermocline position was anomalous regionally, I don't think this strengthens the paper. For one, again, these are the results of a model with coarse gridding in space and time (the authors later point out that such models are to be taken with a grain of salt in areas with complex local bathymetry and local dynamics). More generally, this paper should rely mainly on the observations collected and the MITgcm model. I don't think the cluttered presentation in Fig. 3 really adds anything and it could be simplified with a net positive benefit on presentation.
- 2) Fig 4.
 - a) Fig 4 a— I cannot understand why the data is “averaged to daily values and then low passed” with a 6 d cut off. I do not believe this is necessary... plus if you are going to filter a time series, it is not necessary to average it beforehand. I don't think the filtering actually probably adds that much and I would avoid it.
 - b) This is nice (real, unfiltered data!!) and should be bigger and the main point of this figure. In fact, I think the authors should be comparing IDR and MA with this figure, showing a zoom in like this for both sites. But either way, these are the observations we need to see!
 - c) Ok... I have experience with Nortek instruments in ecosounder mode and I can confidently say that it is not scientifically accepted that the returns in that instrument mode can be simply ascribed to “turbulence” as the authors do

here. This is a contentious claim, and the authors have no other turbulence data that I see to justify it. Moreover, why is this even necessary? Why have a figure about turbulence and then never talk about mixing processes in the text? The authors are mainly interested in the actions of the internal waves in transporting cold (or trapping warm) water to the reef. This doesn't need qualitative hand waving about turbulence, even if the internal waves are important to mixing and mixing is important to bleaching. You just don't have the necessary data to say anything quantitative, so it would be much better to remove this. Finally, I bet you are seeing enhanced scattering from biological sources (zooplankton, etc) that are beings advected up and down by the internal waves. Even if it IS turbulence causing this signal, you can't quantify it in these data so you are better off leaving it out.

- d) Why is tidal elevation here? These internal waves are (probably) baroclinic responses to barotropic tidal forcing. The surface tide will not give you much or any insight to the subsurface internal tide, especially over a hand picked 18 h segment... What is this supposed to show?
- 3) Lines 242-243: "The influence of internal waves, both at tidal and higher frequencies, was far less pronounced during 2019 when the thermocline was depressed due to the IOD." Where is the evidence for this statement? Your time series starts in Dec 2019. Are you saying that there were less internal waves in Dec 2019 than in the rest of the time series? Do you mean that the appearance of sub thermocline water on the reef due to internal waves was less common or absent in 2019? Either way, this statement is a very important one that absolutely needs to be justified if it is going to be in here.
- 4) Fig. 5
 - a) Ok... I have spent a few days trying to puzzle out what exactly is being shown in Fig 5 a (and again in S3, S4a and S5a). This is my best guess: The model is initialised with a density profile taken by the authors on some day in some offshore place. This is used as the initial density profile at all locations in the model. Then the model is run for 15 days. Then (now I am getting into speculation territory) the authors extract the bottom temperature values at all times over the reefs of interest, form some sort of average in time at each near bottom location, then take the temperature at the corresponding depth from the initial, offshore profile and subtract that offshore temperature from the model temperature. Therefore, "red" means that at that depth it is warmer than the offshore initial condition. Is this more or less what is being done?

If so, this needs to be explained somewhere!! The title of the plots is variously referred to as "model mean temperature" and "bottom mean temperature variations." Arguably the quantity you are plotting is not accurately described by either of these titles. It would be more correct to refer to this as an anomaly relative to the initial state or similar.

Second, the CTD cast that you choose to initialise the model is itself simply a snapshot at one time of the evolving vertical position and gradient of the thermocline. Did you test other stratifications? To this reviewer, it would have been a better use of the modelling to compare the internal wave climate

under several different stratifications (i.e. at peak positive 2019 IOD vs “normal” conditions.. These initializations need not be collected by the authors, but instead could come from ARGO or even the model reanalysis. The MITgcm is best used to test the hypothesis that changing stratification impacts internal wave variability at both locations, in addition to exploring how the internal waves are different at the two locations.

Finally, and most importantly, this approach to analyzing the model does not actually tell you anything about the presence or absence of mode-2 internal waves (or any internal waves). These plots need to be re-thought and potentially abandoned entirely. They are confusing and do not speak to what you are trying to assess.

- b) In general, it is confusing to me why there would be so much model shown and so little data in Fig 5 c and d. You have weeks of data... why not show it? I don't think what is being shown here is very convincing in terms of data/model comparison (see next comment).
- 5) The model... I found myself trying to figure out what exactly was done with the model. While the spatial resolution was mentioned, the size of the domain was not. How far into the open ocean does it extend? I couldn't figure out the time step either. I will note that it is somewhat rare to fire up a model and “run it for 15 days” without a spin up period, but perhaps there was one. In general the modelling details were sparse and need to be filled in a bit.

I was likewise a bit confused about the model/data comparison. Yes, this is difficult to do quantitatively, but no attempt has been made here to do so. Instead it is asserted that the model captures the internal wave variability, with a nod to the figures that show wiggly lines from the data and wiggly lines from the model. (I note here that there is –presumably– a typo: these would be depth-AVERAGED velocities, not depth-integrated velocities... or they are plotted in the wrong units). Comparing depth averaged velocities between a model and data is a bit of a funny thing to do if you care about the baroclinic internal wave response, which is not a depth-averaged sort of quantity. So yeah... I'm not sure outside of qualitatively for the depth averaged currents that there is any evidence that the model is recreating the internal wave features in the data.

In S6 both model and a short snapshot of ADCP data in depth and time are shown. But I am confused why the data is only shown for a 2.5 day period, but the full 15 d model time series is shown (see previous comment about fig 5 c and d in the main part of the paper).

- 6) Regarding the “mode-2” structure of the tidal variability at MA.
 - a) First comment is that mode-2 variability is commonly observed and often studied and written about in the literature, so I would remove the statement “relatively rare observation throughout the ocean where attention has largely focused on Mode 1 waves within which the direction of vertical movement of isotherms is consistent throughout the whole water column.” which while

perhaps not strictly incorrect, requires a lot of work by the modifiers “relatively” and “largely.”

- b) You have the data to show whether or not MA and IDR have a different vertical structure of the internal tide. You *do not* need the model for this. Instead you would use the model to diagnose WHY the internal wave structure is different in the different locations. I would suggest an EOF or other modal decomposition to show this from the data.
- c) For the record... internal waves cannot “warm” the water. They can transport heat from one place to another (advective heat flux), and they can give up energy to turbulence, which in turn can lead to mixing, which, then, in the presence of a temperature gradient, can lead to a heat flux (mixing-driven heat flux). What you are showing in the model is that the (I think time mean) vertical temperature profile at MA is different from the initial condition (and IDR) particularly in the mid-reef. You do appear to have some data in fig 4 that shows one example of isotherms separating. I would suggest you lean more on these observations. I do NOT think this paper needs to “discover” that the internal wave climate is complicated over complicated bathymetry, or that the oceanography can be different on different sides of an atoll, even a small one, in order to be an impactful contribution.

Minor comments:

- 1) Lines 182-183: The discussion of SST being a “poor predictor” of thermocline depth and thus the potential that CRW not being good for mesophotic zone corals needs to be more carefully made... the index the authors show in figure 3 b is, in fact, the difference in SSTs between the western and equatorial Indian Ocean. The authors assert that the IOD index captures the anomalous thermocline deepening (which is due most likely to a westward propagating Rossby wave). Nevertheless, after saying that SST as represented in CRW obscures risk to deep corals, in the very next thought they use an SST index as predictive of stress to deep corals. This should be cleaned up.
- 2) Lines 221-222: “resembling at times turbulent bores and, at other times, nonlinear wave trains” Where is the evidence for this? This is speculation and should be removed, it doesn’t add anything.
- 3)

We are grateful to the former reviewer and the new reviewer for taking the time to review our manuscript and provide constructive comments. Please find our detailed responses below.

Reviewers' comments:

Reviewer #1 (Remarks to the Author):

Review of Deep Coral Bleaching Driven by Hidden Changes in Thermocline Depth for Nature Communications

I am now reviewing this revised manuscript for the second time and as previously stated I am impressed with the physical oceanography and its causal role in the observed bleaching of scleractinian corals on MCEs in the Indian Ocean. I also appreciate the authors response to the previous reviewer's comments. Nonetheless, I still believe there are issues that should be addressed.

- Thank you for accepting to review this manuscript again, we are grateful for the constructive comments.

Line 18; Suggest this reads as follows; "Our results demonstrate the potential vulnerability..."

- We have edited the sentence as suggested.

Line 26; Here, and throughout the manuscript "Scleractinian" is not capitalized. It is not a taxon and should be in lower case.

- We have edited the words as suggested.

Line 27; Suggest this reads as follows; "...sea surface temperatures (SSTs) and subsequent coral bleaching; current estimates..."

- We have edited the sentence as suggested.

Line 32; Suggest this reads as follows; "...depths of 30-150 m and have been estimated to occupy..."

- We have edited the sentence as suggested.

Line 33-34; Suggest this reads as follows; "It has been hypothesized that these communities are buffered against anthropogenic changes, and therefore might represent a refuge that replenishes shallow-water coral species with vertically dispersed larvae^{8,10}. Or words to that effect....."

- We have edited the sentence as suggested.

Line 51; "in-situ" is not hyphenated and should be italicized as it is Latin.

- We have edited the word.

Line 53; Per my previous review. The fact that a flawed metric (i.e., DHW) is being used by a large proportion of the community is not a justification for its continued use. If the authors insist on using this metric, knowing these flaws as I pointed out in my last review, then they must at least tell the community that there are known problems, with references, especially as it relates to its predictive capabilities.

- More details have been added to the text, including reference to the literature to highlight potential flaws with this metric. Furthermore, in our previous revisions to the manuscript we added text and data to demonstrate that the NOAA CRW predictions were supported by the modelled local temperature fluctuations shown in supplementary figure S2, and *in situ* temperature data recorded by the CTD mounted on the ROV (supplementary figure S1), which both indicate no thermal stress in shallow waters in November 2019.

Line 63; Reviewer #2 had some keen insights on this, and I will be interested in seeing how the authors deal with the issue of post-bleaching coral communities being dominated by “winners” as a reason for little to no bleaching on shallow reefs compared to potentially more thermally sensitive corals at MCE depths. The recovery you discussed in response to the reviewer, in the absence of any seascape genetic data, was likely a result of recruitment from local populations, or the winners so likely more resistant to thermal stress. Also, 3000 corals in 2019 and 1000 in 2020 on shallow reefs- what happened here?

- In our last revision of the manuscript, we added some additional information to the manuscript in response to the reviewers' comments regarding the presence of 'thermally-resistant colonies in shallow water' to support our conclusions that lack of bleaching in shallow waters was more likely a result of no thermal stress rather than a lack of colonies observed.
- From line 132, we state that “Shallow-water reefs in the Chagos Archipelago may now be composed of a higher proportion of heat-tolerant genera and species of scleractinian corals than before the mass bleaching events of 2014-2017, which is possibly reflected by the low prevalence of bleaching recorded on shallow-water reefs in the current study [...]. The water temperature during all other dives remained below 29.5°C for the duration of the dives. Thus, the lack of bleaching observed on shallow-water reefs (15-20 m) is likely due to an absence of thermal stress rather than a lack of coral colonies on the reef and/or a high proportion of thermally tolerant individuals [...]. On shallow-water reefs, high bleaching variability at local scales has been found to be partly due to differences in species composition in terms of both taxa and abundances. In this study, significant differences in coral community composition were observed between depth bands within each site [...]. At mesophotic depths (from 60 m to 90 m), no significant differences in coral community composition were detected between the two sites [...]. Furthermore, the few coral species that span the depth distribution from shallow to upper-mesophotic waters (30-40 m) down to the lower mesophotic zone (60-70 m, and 80-90 m for *Leptoseris spp.*) experienced more severe bleaching in terms of prevalence and severity at mesophotic depths, with more coral bleaching observed at MA compared to IDR (supplementary table S4), indicating that coral composition is not the only driver of differences in coral bleaching along the depth gradient.”
- We agree that mesophotic corals may also be more thermally-sensitive, and our data demonstrate a temperature difference of 3-7°C above the normal seasonal range at

mesophotic depths, unlike in the shallow waters, where no thermal stress was detected at the time of our survey (from *in situ* measurements and satellite derived predictions).

- The difference between the number of corals assessed between 2019 and 2020 is because we had differing number of images surveyed between 2019 and 2020. In March 2020, we were unable to survey all locations due to poor weather conditions and the Covid pandemic cutting short the second cruise. In total in 2019, we surveyed 150 images in the 15-20m depth zone, while we surveyed only 30 images in March 2020 (see Table S1, supplementary material).

Figure 1. The euphotic zone, classically defined in the oceanographic community is 0-200 m and the mesopelagic zone is 200-1000 m. The MCE depth definition is placed, or should be placed, over these long accepted oceanographic depth descriptions as near coastal benthic habitat.

- Thank you for highlighting the discrepancies here. The categories in Figure 1 should refer only to the benthic zones rather than oceanographic depth descriptions. This has been clarified in the figure legend.

Line 110; "5" should be "five"

- We have edited the word, thank you.

Line 128-135; There it is-the "winners" argument, but then you try and support your interpretation with a flawed metric. Again, if you are going to do this you must state clearly for the readers the flawed aspects of the metric.

- Thank you for highlighting this. We have added specifics to the text to highlight that this metric has limitations. In our previous revisions to the manuscript, we added additional text and data to support the predictions made by the NOAA CRW that there was no thermal stress in shallow water in November 2019 (we support our interpretations with temperature data collected in the field from the ROV in 2019).

Figure 2; Need a fuller description of statistics. As requested in my previous review what primary analysis, and post hoc analysis, was run? What were the DF, F-value, p value corrections for multiple comparisons, etc. as applicable.

- A full description of the statistics was provided in the methods section of the manuscript previously submitted. However, detailed results were missing from the main text in the results section and the figure legend. We apologise for this oversight, and we have added specific details to the results text, figure legend and detailed results in supplementary tables S5 & S6.

Lines 155 and elsewhere; ANOSIM provides a test to detect differences in two or more groups of "sampling units" based on ranked similarities. First, why not present the results as an NMDS or cluster plot? Would be much easier for the reader to see the depth comparisons of coral communities between depths and sites. Did the authors ensure that the heterogeneity of dispersion, which ANOSIM is very sensitive to, is equivalent or at least similar between depths and sites? Also, why the

more permissive p value of 0.1 to start with? Is it possible that at a minimum p value of 0.05 some of these comparisons would not have been significant?

- We didn't originally show the MDS plot to save space within the manuscript and we felt it was appropriate just to show the statistical results. However, we have added the MDS plot to the supplementary materials (based on the PERMANOVA analysis described below).

- We have re-analysed our scleractinian coral community data using PERMANOVA, with a balanced design (please see results in the supplementary materials). There is significant heterogeneity of variance among some of the groups of sites/depths, but the largest differences are between communities at different depths rather than between two sites at the same depth (see boxplot and PERMDISP results in table below). PERMANOVA with a balanced design is robust to the issues of heterogeneity of variances among groups. Furthermore, we are mainly interested in the variance in coral community structure between the two sites at the same depth. The PERMDISP results and boxplot below show significant homogeneity of variance for Ile des Rats and Manta Alley at 60m and 80m, with significant heterogeneity of variance between the two sites at 15m and 30m (although the t -value is fairly low here). Thus, the PERMANOVA results can be considered robust.

Groups	t	P(perm)
(IDR-15,IDR-30)	5.3373	0.0010
(IDR-15,IDR-60)	5.9659	0.0010
(IDR-15,IDR-80)	10.789	0.0010
(IDR-15,MA-15)	4.1261	0.0010
(IDR-15,MA-30)	3.5192	0.0030
(IDR-15,MA-60)	4.0816	0.0010
(IDR-15,MA-80)	12.081	0.0010
(IDR-30,IDR-60)	10.54	0.0010
(IDR-30,IDR-80)	15.025	0.0010
(IDR-30,MA-15)	0.74318	0.4860
(IDR-30,MA-30)	6.8273	0.0010
(IDR-30,MA-60)	8.7443	0.0010
(IDR-30,MA-80)	16.723	0.0010
(IDR-60,IDR-80)	4.7789	0.0010
(IDR-60,MA-15)	9.2125	0.0010
(IDR-60,MA-30)	0.81446	0.4900
(IDR-60,MA-60)	1.782	0.1330
(IDR-60,MA-80)	5.4262	0.0010
(IDR-80,MA-15)	13.599	0.0010
(IDR-80,MA-30)	4.593	0.0010
(IDR-80,MA-60)	6.517	0.0010
(IDR-80,MA-80)	0.3041	0.7740
(MA-15,MA-30)	6.0994	0.0010
(MA-15,MA-60)	7.4954	0.0010
(MA-15,MA-80)	14.999	0.0010
(MA-30,MA-60)	0.55073	0.6050
(MA-30,MA-80)	5.0361	0.0010
(MA-60,MA-80)	7.3036	0.0010

- The PERMANOVA results indicate significant differences in the coral communities between sites and depths. We would expect differences between depths as the environmental conditions change over the depth gradient, and thus, so do the benthic communities. However, the differences between sites at the same depth are minimal (see overlap of data points on the MDS plot above and the low t-values from the PERMANOVA analysis, Table S2-1, and the number of shared coral species between sites at the same depth, Table S3). Thus, the significant difference in coral community composition at the two sites at the same depths is likely driven by the presence of a low number of rare coral species. Thus, we can be confident that the difference in bleaching between sites at the same depth is not due to large differences in the coral species present at the two sites as these are largely the same, aside from a few rare species.
- The p-value presented in the previous version of the manuscript for the ANOSIM results is on the contrary less permissive, as 0.1% is equivalent to 0.001. However, this is now irrelevant as we are using PERMANOVA instead of ANOSIM and the results text has been updated accordingly.

Line 165; "spp." is not italicized.

- It has been edited, thank you.

Line 250; Is this supposed to say; "(See modelling methods)"?

- Indeed, it has been edited, thank you.

Line 283; Suggest this sentence reads as follows; "In summary, our results show that scleractinian corals at mesophotic depths from the Chagos Archipelago in the Indian Ocean are susceptible to thermal stress, and bleach in a similar manner to that observed for shallow-water corals in contrast to other studies on mesophotic corals from different geographic areas²⁶, 32-34."

- We have edited the sentence as suggested.

Line 286; Suggest this reads as follows; "Our study adds more evidence that MCEs are not, a priori, a universal source of coral recruits for replenishing shallow water coral reefs^{12, 13, 37-39."}

- We have edited the sentence as suggested.

Lines 334-338; So, transects were employed. How long were they? How many images were taken at each depth? Were the same number of images taken at both sites at both times? What was the angle of the oblique imagery?

- As described previously, linear transects were not employed due to the nature of the currents around Egmont Atoll preventing the ROV from operating along a linear line. However, a set number of still images of the seabed were collected from each depth zone specified (90 images per depth). The number of images taken per site/depth has been added to the supplementary table S1, for clarification.

Lines 339-342; The statement that non-orthogonal, or oblique, imagery is taken by the deep-sea communities is true. But they usually collect those images, along with scale (!), and analyze them a priori knowing that different techniques (i.e., multiple lasers, Canadian grids) will be needed to obtain orthorectified images and accurate community characterizations for multiple ecological metrics such as abundance and percent cover. Regardless of whether plot or plotless designs are incorporated the imagery must be orthorectified.

- As mentioned previously, we experienced significant issues in piloting the ROV in difficult currents and uneven topography around Egmont Atoll. In addition, the lasers on the SD camera broke prior to the first surveys being undertaken. We were not able to repair the lasers and as we were two days steam from the nearest port, we had to continue with our data collection without them. We fully understand the need to collect images along with a measure of scale to standardise data collection and subsequent analysis. Given our predicament in a remote location, the needs of other researchers on the cruise and the funds already spent on ship time and transporting equipment to the ship, we continued with

our surveys and standardised image collection from the seabed as much as was possible. As detailed in the manuscript, the angle of the camera was kept in the same position and we maintained a consistent altitude above the seabed when taking the still image. We have been as transparent as we can about how the data were collected, and we have added an additional information to the manuscript to ensure the reader is fully aware of the circumstances of the data collection. Although we are unable to quantify bleaching over a given area, we believe our standardised method of image collection enables us to compare bleaching prevalence and severity within our study using assessments of individual colonies (similar to McClanahan et al., 2001). We believe that the technical elements have no material impact on the validity of the findings of the manuscript.

Line 347; Why were the depth bands discontinuous?

- The depth bands were discontinuous to align with the specific depths previously surveyed in the literature. Furthermore, we were taking part in a multidisciplinary cruise and time had to be allocated equally amongst researchers. Thus, we had to maximise our time in the water and chose to target discrete depth bands rather than undertake continuous surveys across the entire depth gradient.

Lines 362-372; You only present the “community” analysis on zooxanthellate corals for the presence/absence data. The additional information is unnecessary. Amend it for the analysis presented in the manuscript.

- We have edited the text as suggested.

Line 386; “coral colour” is actually a function of its reflectance captured by the image under a specific light field. Without a reference, a control, based on a simultaneous image of a Lambertian surface you cannot accurately and consistently quantify image to image difference in “coral colour” even with a known light source. More on this later.

- We appreciate this and do not deny that image analysis of bleached coral should include reference to a standard material. However, we are unable to return to the field to photograph a reference material. At the time of the survey, our aim was to collect baseline data on the diversity and distribution of benthic communities from shallow to mesophotic depths. We did not anticipate observing bleaching on the reefs and we did not go prepared to survey bleaching. However, the presence of bleaching on the reefs on our arrival offered an unprecedented opportunity to document bleaching at mesophotic depths in parallel to oceanographic data being collected at the same time for the same site. We have a number of images of bleached corals at different depths that were subsequently sampled for genetic analysis and thus, were verified as bleached onboard the ship (images also available). The presence of the bleached colonies across the depth zones was striking and the bleached colonies were frequently visible from 15-20 m above the seabed during our descent with the ROV. We have different bleaching severities within the same image (thus exposed to the same light conditions). Images for analysis were selected only if they had even lighting and

the presence of video data meant we could review the colonies from multiple angles using the video footage to further verify the presence of bleaching.

Lines 414-418; As stated previously; "Also, there is no indication of how bleaching prevalence was actually calculated because to compare between sites one would have to/should know the aerial extent of the survey. That is the percentage of bleached coral at site A was 50% and site B was 75% based on a 100 m² survey. Or better yet do transects at each depth, even plotless transects would be fine as long as the areas surveyed were the same. Just like Quimpo et al. 2020, the paper the authors reference, did in their paper.

- Methodological detail on how images were collected across survey sites was provided in the methodology and the specific number of images per site and depth has been added to the supplementary table S1. While we are unable to quantify the area surveyed, we believe that assessing individual colonies for bleaching and determining the proportion of colonies bleached is an adequate measure for the study described here. Assessing individual colonies on a reef rather than surveying a given area of reef has been undertaken previously (e.g McClanahan et al., 2001). While we understand the survey design is not perfect, due to the reasons described above, we still believe that the data tell a compelling story and should be shared with the research community. We have been transparent in how we collected the data in the manuscript.

Response to authors comments of prior review

I appreciate the efforts of the authors to examine both irradiance and cloud cover at their sites as it relates to coral bleaching. I also understand that corals at mesophotic depths are probably naïve to temperature increases of 3-7°C. Figure C1 is helpful although changing the scale to allow a closer examination of the changes in irradiance at mesophotic depths would have been more informative. However, the authors do provide calculations of optical depths for the irradiance data they do have that are based on calculated Kd. The authors state that only 1% of surface PAR reaches 90 m so irradiance is not an issue as it relates to bleaching. But the corals at these depths are acclimatized to these extremely low irradiances. Any change in the optical properties may allow increased water transparency and increased irradiances, albeit small when thinking in terms of shallow coral reefs, but not inconsequential for extreme low-light acclimatized corals. For instance, the Kd for the 1% OD of 63 m is 0.073 while for 97.8 m the Kd is 0.047 from your table. These are not inconsequential differences and suggest significant changes in water transparency which could have consequences related to the role of light in bleaching at mesophotic depths, which on shallow reefs is primarily a light driven phenomenon; temperature sets the table, but light drives the process. The authors should give the reader the data and a chance to assess. Provide the OD data, give an example and say it is possible that these oceanographic processes, especially internal waves, could change the transparency of the water column and could have contributed to the bleaching process.

- We have added the table of light optical depth, which we provided in our previous response to your comments, to the supplementary materials file (table S7), as well as specific details in the manuscript text.

- Any comment on the impact of internal waves on water clarity would be entirely speculative and also assumes detailed knowledge and understanding of the temporal scales over which light may be attenuated (presumably by elevated resuspension) and the response of the coral to this. We recognise that this may be an interesting consideration within a future study but is beyond the scope of the present paper.

On the issue of using coral color as an index of bleaching extent and severity. As suggested above again, the calculation of a bleaching index based on coral color requires that all colonies being assessed in the image be orthogonal, and at the same height above the substratum so that the light source illuminates the coral evenly and at the same intensity. If this does not occur then this index, basically a measure of coral reflectance, requires, for each image, a reference to compare against. The reference material must be Lambertian in nature; a commonly used material is Spectralon, which can provide a diffuse reflection that is equal across all visible wavelengths. This allows the user to compare the reflectance of the standard to the reflectance of the subject under the same light source (and any differences associated with light intensity). For images that are not orthogonal such as used here multiple references would be needed since the distance from the light to the subject varies, and therefore so does light intensity across the image. This can be all entered in the authors BIIGLE program for appropriate annotation and is essential to compare coral bleaching between sites, and between years.

- We appreciate this and do not deny that image analysis requires reference to a standard material. However, we are unable to return to the field to photograph a reference materials and add them to Biigle. Biigle only allows adjustment of contrast, brightness, hue and saturation, thus our images in Biigle cannot be corrected for colour. We could apply a colour correction to all our images outside of Biigle, however, we would need to seek additional support and advice on how to undertake this. However, we believe that the bleached colonies were very obvious on the reef and have verified bleached colonies viewed at multiple depths on camera with coral fragments brought to the surface.
- At the time of the survey, our aim was to collect baseline data on the diversity and distribution of benthic communities from shallow to mesophotic depths. We did not anticipate observing bleaching on the reefs and we did not go prepared to survey bleaching. However, the presence of bleaching on the reefs on our arrival offered an unprecedented opportunity to document bleaching at mesophotic depths in parallel to oceanographic data being collected at the same time for the same site. While we are unable to photograph reference material, we have a number of images of bleached coral at different depths that were subsequently sampled for genetic analysis and thus, were verified as bleached onboard the ship (images also available). The presence of the bleached colonies across the depth zones was striking and the bleached colonies were frequently visible from 15-20 m above the seabed during our decent with the ROV. While our data were not collected under a perfect experimental design, we believe the story is extremely valuable and should be shared with the research community.

As far as bleaching prevalence I still believe that to compare between depths or sites one would have to/should know the aerial extent of the survey. That is the percentage of bleached coral at site A was

50% and site B was 75% based on a 100 m² survey. The authors describe taking images, and the number of images, but do not adequately describe the process-see comments above for Lines 334-338.

- Methodological detail on how images were collected across survey sites was provided in the methodology and the specific number of images per site and depth has been added to supplementary Table S1. While we are unable to quantify the area surveyed, we believe that assessing individual colonies for bleaching and determining the proportion of colonies bleached is an adequate measure for the study described here. Assessing individual colonies on a reef rather than surveying a given area of reef has been undertaken previously (e.g. McClanahan et al 2001). While we understand the survey design is not perfect, due to the reasons described above, we still believe that the data tell a compelling story and should be shared with the research community. We have been transparent in how we collected the data in the manuscript.

The authors contend that using the number of individuals to assess species diversity is a valid approach as described by Gotelli and Cowell 2001. Their techniques utilize rarefaction curves and I see no description of that in the M+M of this manuscript. Also, my reading of Gotelli and Colwell 2002 suggest that using species density, versus species diversity/richness would be a more appropriate metric to derive for quantitative studies, and therefore requires a knowledge of area analyzed. This is best accomplished using orthogonal images.

- The reference to Gotelli and Colwell 2001 was made in our previous response to the reviewer comments on assessing species diversity from images of the seabed – the reviewer comment was made in reference to our example image from Biigle showing our annotations. We discussed in our previous reply that individual-based rarefaction can be used to enable direct comparison of species diversity among sites/depths based on the number of individuals encountered, thus we do not need to know the area of the image being annotated. We have not assessed species diversity using rarefaction curves within this manuscript; thus details are not provided on this in the M&M.

Again, the physical oceanography is amazing! The authors should couple that data to the bleaching prevalence data alone, once the M+M for the bleaching prevalence data is better explained, and omit everything else.

- Thank you for your comments. We have improved our explanation of the bleaching prevalence and severity data and how these were determined, and we have been completely transparent about the limitations of our data.

Reviewer #3

I was brought in to review the physical oceanography reported in this manuscript. I have expertise in studying the dynamics of internal waves in the ocean using observations similar to those reported here.

- We appreciate the thorough review from a specialist in the oceanographic element of this study.

In general, there is a reasonably detailed set of measurements of the physical state of the ocean at the two study sites. These observations come from moorings with temperature, salinity, and currents. Based on the description in the methods, the mooring design and the data that the moorings collected seems appropriate to assess the variability of temperature over the upper ~100 m with good vertical resolution on frequencies from minutes to weeks. Having this data is very good and an absolute prerequisite here. Unfortunately, however, the mooring time series start after (dec 2020) the primary bleaching period (nov 2020), and the data for whatever reason is given second billing to two numerical models.

- We appreciate the reviewer's comment regarding the start of observations being in December 2019, after the bleaching was observed during November of the same year. However, we did in fact deploy moorings for 2-3 weeks during the November cruise but did not include them in this paper because the data were almost entirely from within the SML due to the extremely deep thermocline at the time. The temperature data from November thus demonstrate a vertically homogenous surface layer to a depth of 60 m, which is the limit of the moorings at this site due to the steep slopes below this depth that prohibit the deployment of moorings other than at a couple of sites that we identified from the multibeam survey. It is for this reason that the MITgcm simulations referred to below were so important, in filling in the unavoidable gaps in the observations and providing an assessment of how the internal waves potentially impact the thermal regime at depth, below the depths for which we have observational evidence.

MITgcm is used by the authors to model the internal wave variability. It is initialised by a CTD profile the authors collected and forced by the semidiurnal and diurnal barotropic tides. The model is (sort of) compared to the data, and then the authors create a number of plots of model output that they use to argue about differences between the two reef sites. The authors settle on "mode 2" internal waves as differing between the two sites. I am fairly sceptical that the analysis of the model in the way presented here supports that conclusion to the exclusion of other hypotheses. I would have preferred to see more analysis of the data, and a deeper dive into the model insight about how the internal wave climate is different between the two sites.

- We agree with the reviewer that these environments are extremely complicated and other factors may play a role; however, given the challenge of obtaining observation data from this environment and the need to explain the observed bleaching characteristics, we have prioritised the most plausible mechanism for which we have observational and modelling evidence. This is not meant to exclude other mechanisms that would require (prohibitively complex) additional observations and we trust we have not been too bold in our language in suggesting that other mechanisms are not to be excluded. We have checked our language to this effect but we also believe we have been thorough in our assessment of the role played by the mode 2 waves through the modal decomposition that the reviewer may have missed as it's at the end of the Supplementary Material (as we felt it would be too technical and would distract from the main story for the wider audience that is likely to be more biologically-orientated).

It should be understood that my comments that follow are in an effort to get the best out of these data, and that I don't believe that the manuscript should be rejected due to the PO

component of the paper. That said, I have a number of fairly significant concerns with the presentation and interpretation.

For the record, I actually do not believe that “definitively” identifying the presence of some internal wave pattern or another is a precondition for this being a valuable contribution. Simply identifying, in the data first, and the model second, that the different reefs have different internal wave climates is probably enough to POSE the hypothesis that this is the reason why one bleach and the other did not. More data/modelling would be needed to definitely show this.

- We understand the reviewer’s concerns and attempted to write the paper without being too bullish about the certainty of our results. As referred to above, we have tried to be transparent regarding the uncertainties but we also consider that the observations, obtained from an enormously challenging location due to bottom slope steepness, and numerical modelling represents compelling evidence for our interpretation.

Finally, there has been some fairly strident back-and-forth between the authors and the first two reviewers about the value of SST and “coarse” numerical models in predicting bleaching risk in mesophotic corals. My take is the following: of course a mode-derived “early warning” product tuned to the ocean surface mixed layer and shallow (<30 m depth) corals would have questionable application to 40-80 m depths. Is there actually debate about this?

- Our key contribution to this aspect of the study is that there is very little argument to this (i.e. mesophotic reefs) because it’s simply not yet received the same level attention of shallow reefs whilst, also, the dynamics responsible for controlling the thermal environment are completely different.

*As the authors point out, because of the importance of internal wave variability, any global model driven early warning system is likely to be quite challenged. (As an aside, I will note that global models like mercator do not get the internal waves correct *anywhere* and so by definition can not at present capture this impact for any depth). Simply pointing out that current generation “bleaching warning systems” are not going to be as effective for mesophotic corals because of internal wave variability, without getting into the “difficulties” of measuring the position of the thermocline – which is actually not that hard and is regularly done with moorings all over the world – is plenty of a conclusion enough in that realm for a paper like this.*

- Yes, we agree that we could limit our findings to this but we would then be missing the point that those interested in bleaching of mesophotic coral reefs need to consider using higher resolution, internal wave-resolving models than currently offered by the global models.

I hope the authors find these comments useful.

- We are very grateful to have the perspective of an oceanographer in reviewing this paper and we trust that the reviewer appreciates the time we have put into our response to their review that we are grateful to have received as we look to continually improve the paper.

I have written many of them in what follows.

It is not my position that every suggestion has to be followed up on. But I do worry that at present the PO story is incomplete and in some ways confusing and potentially even wrong.

We want to get as much as that out of the paper as possible before it is published.

Major comments about the PO:

1) The presentation in Fig. 3 is confusing. First of all, Fig 3 is from a model reanalysis... It is not data, it is the output of a model. The model does assimilate data, but it needs to be clearly understood that these are not actual observations, including within the figure/caption itself.

- We were not under the impression that we suggested the figure was developed from data rather than model output but have checked to ensure that this is the case. We have however included explicit reference to the use of the CMEMS model output in generating the figure to avoid any misunderstanding.

I have comments below about each subpanel.

a) There really is no need to show the entire reanalysis period. Why is this done? I much prefer the presentation in Fig. S2. Fig 3 a and b are far too cluttered and this story is about 2019-2020, not the decade beforehand. I strongly suggest that this be re thought. It is almost impossible to understand anything about the temperature fluctuations in fig 3 a.

- We can certainly understand the reviewer's perspective on this but we wished to use the opportunity to demonstrate how anomalous 2019 was within the context of recent history, for which presenting the thermocline depth as far back as 1994 is helpful because one can see the deep thermocline in 1998 and again 2006. We agree that this makes the detail more difficult to discern but that is not the purpose of this panel.

b) Fig 3 b should be zoomed into a much tighter time range. Again, there is little value in showing a long time series, since the point of the paper seems to be that 1) regional oceanography was anomalous (this has been shown by others and comes with a few references in the text) and 2) local effects matter. Neither fig 3 a or 3 b needs a decadal perspective.

- We appreciate the reviewer's concerns but, as with the point above, these two panels provide the temporal context to how anomalous 2019 was; whilst others have shown that the regional oceanography was anomalous, none have (as far as we are aware) done so within the context of thermocline depth. Given the critical role played by the thermocline in setting the depth range over which internal waves may exert an influence, it is important to demonstrate the time evolution of thermocline depth over a long period to highlight where 2019 sits within the context of the ocean state over the past 2 -3 decades.

c) Fig 3 c d and e are unnecessary. While somewhat interesting that the thermocline position was anomalous regionally, I don't think this strengthens the paper. For one, again, these are the results of a model with coarse gridding in space and time (the authors later point out that such models are to be taken with a grain of salt in areas with complex local bathymetry and local dynamics).

- These panels demonstrate the thermocline depth over the Indian Ocean, for which the global model output is much more robust than around the regions of complex bathymetry. The importance of these panels lies in demonstrating the spatial extent of this thermocline evolution over the Indian Ocean. We were acutely aware of the importance of demonstrating that our results weren't a special case that had limited applicability elsewhere. These panels demonstrate that the thermocline deepening due to the IOD extended throughout the central, equatorial and western Indian Ocean, thereby influencing the MCE in locations such as La Reunion, Mauritius, Seychelles and the Maldives. We trust

that the reviewer agrees that making it clear how this basin-scale evolution of the thermocline impacts multiple locations is important to bring to the reader's attention.

More generally, this paper should rely mainly on the observations collected and the MITgcm model. I don't think the cluttered presentation in Fig. 3 really adds anything and it could be simplified with a net positive benefit on presentation.

- We appreciate the reviewer's perspective but hope that our explanation above is accepted.

2) Fig 4.

a) Fig 4 a— I cannot understand why the data is “averaged to daily values and then low passed” with a 6 d cut off. I do not believe this is necessary... plus if you are going to filter a time series, it is not necessary to average it beforehand. I don't think the filtering actually probably adds that much and I would avoid it.

- The daily averaging was an initial step taken to enable an assessment of the data before the filtering was decided upon for the purpose of the presentation in Figure 4a. We agree that the explanation is perhaps confusing but would stress that the 6 day filtering is necessary for the purpose of presentation; without it, the tidal influence renders the long-term evolution much more difficult to discern because of the higher frequency variability present in the data. Please note that the key finding from this panel is that, in support of the indications offered by the global model output, the thermocline was below the depth of observations during November 2019 (this why we don't include moored observations from that cruise) and then shoaled to more typical depths up the second cruise in March 2020 when the coral demonstrated good recovery.

b) This is nice (real, unfiltered data!!) and should be bigger and the main point of this figure. In fact, I think the authors should be comparing IDR and MA with this figure, showing a zoom in like this for both sites. But either way, these are the observations we need to see!

- We agree that a direct comparison could be useful but also recognise the limitations on space and the rather subtle differences in the equivalent data from IDR and MA. For this reason the figure requested by the reviewer is in fact already in the paper but within the Supplementary material in Figure S7 (where we trust the reviewer agrees, the differences are subtle and more likely distracting to include the main text). We would thus prefer to retain the presentation of Figure 4 as it currently is because the panels demonstrate a) the local evolution (shoaling) of the thermocline suggested by the basin-scale model analysis presented in Figure 3, b) the tidal periodicity in thermocline behaviour over the slope at MA and c) the detailed structure of individual tidal excursions at that location.

c) Ok... I have experience with Nortek instruments in ecosounder mode and I can confidently say that it is not scientifically accepted that the returns in that instrument mode can be simply ascribed to “turbulence” as the authors do here. This is a contentious claim, and the authors have no other turbulence data that I see to justify it. Moreover, why is this even necessary? Why have a figure about turbulence and then never talk about mixing processes in the text? The authors are mainly interested in the actions of the internal waves in transporting cold (or trapping warm) water to the reef. This doesn't need qualitative hand waving about turbulence, even if the internal waves are

important to mixing and mixing is important to bleaching. You just don't have the necessary data to say anything quantitative, so it would be much better to remove this. Finally, I bet you are seeing enhanced scattering from biological sources (zooplankton, etc) that are beings advected up and down by the internal waves. Even if it IS turbulence causing this signal, you can't quantify it in these data so you are better off leaving it out.

- We agree that the echosounder certainly doesn't represent turbulence and have revised the text to moderate our reference to turbulence. We did actually measure turbulence directly with our MSS profiler during our last cruise in March 2022 and the high scattering layers in the echosounder are turbulent (but could also include biological scatterers) but we appreciate that this analysis isn't (and won't be because it's beyond the scope as the reviewer correctly points out) included in the current paper.

d) Why is tidal elevation here? These internal waves are (probably) baroclinic responses to barotropic tidal forcing. The surface tide will not give you much or any insight to the subsurface internal tide, especially over a hand picked 18 h segment... What is this supposed to show?

- The elevation was included at a late stage in the manuscript preparation to provide a reference for the interpretation of currents depicted in Figure 4c. The instantaneous currents are phase-locked to the barotropic tide because the baroclinic response is local rather than associated with remotely-generated internal waves that then break upon the slopes surrounding Egmont. This a good point of the reviewer because we are acutely aware that the archipelago as a whole is inevitably going to be an effective generator of internal waves that radiate away into the surrounding ocean; we obviously cannot robustly include these effects (because the GEBCO bathymetry we'd to use completely misrepresents the slope angles) in the modelling despite their potential role which is why we explain this possible role towards the end of the paper.

3) Lines 242-243: "The influence of internal waves, both at tidal and higher frequencies, was far less pronounced during 2019 when the thermocline was depressed due to the IOD." Where is the evidence for this statement?

- Apologies that this isn't clear in the text and we have modified the MS accordingly. We were referring to the impact of internal waves in MA (so at a depth of 60-70 m) during November 2019 when the thermocline was at 100 m. As a result, MA was inundated with warm surface waters and was vertically homogenous, so internal waves were unable to exist at those depths until the thermocline recovered to more typically depths during early 2020.

Your time series starts in Dec2019. Are you saying that there were less internal waves in Dec 2019 than in the rest of the time series? Do you mean that the appearance of sub thermocline water on the reef due to internal waves was less common or absent in 2019? Either way, this statement is a very important one that absolutely needs to be justified if it is going to be in here.

- As stated above, our observations started in November 2019 but almost all of the moored measurements were within the upper 100 m and thus failed to capture the deeper internal waves. The only exception was a mooring that (fortuitously!) sank after sliding down the very steep slope at IDR during November 2019. This is used to compare with the numerical modelling output in Figure S6. We have revised the text to clarify that the November

conditions meant cold water didn't reach the shallow reefs although we trust the reviewer appreciates that this paper is specifically concerned with the mesophotic reef rather than the shallow reefs.

4) Fig. 5

a) *Ok... I have spent a few days trying to puzzle out what exactly is being shown in Fig 5 a (and again in S3, S4a and S5a). This is my best guess: The model is initialised with a density profile taken by the authors on some day in some offshore place.*

- Correct; the model is initialised with a deep CTD profile taken from the channel to the north of Egmont and where, we assume, the local influence of internal waves is minimised such that the CTD profile is representative of unperturbed conditions.

This is used as the initial density profile at all locations in the model.

- Correct

Then the model is run for 15 days.

- Correct

Then (now I am getting into speculation territory) the authors extract the bottom temperature values at all times over the reefs of interest, form some sort of average in time at each near bottom location, then take the temperature at the corresponding depth from the initial, offshore profile and subtract that offshore temperature from the model temperature.

- Correct, although we would suggest referring to the offshore profile as an 'unperturbed' profile

Therefore, "red" means that at that depth it is warmer than the offshore initial condition.

- Correct – red means that the seabed is experiencing, in a time-averaged sense, warmer conditions than would be found in the absence of the internal waves.

Is this more or less what is being done?

- Yes – perfect!

If so, this needs to be explained somewhere!! The title of the plots is variously referred to as "model mean temperature" and "bottom mean temperature variations." Arguably the quantity you are plotting is not accurately described by either of these titles. It would be more correct to refer to this as an anomaly relative to the initial state or similar.

- We agree that this needs to be clarified and we have carefully examined the text to ensure that our approach is understandable.

Second, the CTD cast that you choose to initialise the model is itself simply a snapshot at one time of the evolving vertical position and gradient of the thermocline.

- To an extent; it was indicative of the unperturbed (in the sense of internal evolution over the slopes) environment during November 2019 when the bleaching occurred. So, yes it was a snapshot but it was a snapshot at the time of interest when the bleaching needed to be explained.

Did you test other stratifications?

- Yes, we ran the model with the stratification for both November 2019 and March 2020. The two scenarios are presented in Figure 5 (MA, March 2020) and S3 (IDR, March 2020) and for November 2019, S4 (MA) and S5 (IDR)

To this reviewer, it would have been a better use of the modelling to compare the internal wave climate under several different stratifications (i.e. at peak positive 2019 IOD vs “normal” conditions..

- As stated above, this is precisely what we did and trust that the reviewer agrees our approach was therefore correct.

These initializations need not be collected by the authors, but instead could come from ARGO or even the model reanalysis. The MITgcm is best used to test the hypothesis that changing stratification impacts internal wave variability at both locations, in addition to exploring how the internal waves are different at the two locations. Finally, and most importantly, this approach to analyzing the model does not actually tell you anything about the presence or absence of mode-2 internal waves (or any internal waves). These plots need to be re-thought and potentially abandoned entirely. They are confusing and do not speak to what you are trying to assess.

- As stated towards the end of the Methods (lines 503-521), the modal decomposition was performed to identify the role of Mode 2 waves. The point of this analysis and the presentation of Figure 5 is to demonstrate that the internal wave exacerbates the heating experienced at depths where the bleaching was observed, i.e, the red shading in Figure 5a. Comparing the panel with Figure S3 which shows the equivalent impact at IDR where no bleaching was observed, we can see an absence of red shading because the internal waves have a minimal impact on the near-bed thermal regime. So the point of Figure 5 is to show the reader that internal waves further enhance warming and, most importantly, the absence of an equivalent effect at a nearby location at the same island highlights the fine-scale approach one must take in understanding bleaching throughout the MCE.

b) In general, it is confusing to me why there would be so much model shown and so little data in Fig 5 c and d. You have weeks of data... why not show it? I don't think what is being shown here is very convincing in terms of data/model comparison (see next comment).

- There are gaps in what the observations show, i.e. the deep thermocline in November 2019 meant all moorings, that couldn't be deployed any deeper because the slopes were too steep (as experienced on the Nov. 2019 cruise when we tried and the mooring slipped 100 m down-slope) were within the SML. Furthermore, the moorings provide high temporal and vertical resolution at a couple of fixed locations, limiting the evaluation of how the internal

waves influence the wider slope region. This is why the model output is so important and we trust we have struck a good balance.

5) The model... I found myself trying to figure out what exactly was done with the model. While the spatial resolution was mentioned, the size of the domain was not. How far into the open ocean does it extend? I couldn't figure out the time step either. I will note that it is somewhat rare to fire up a model and "run it for 15 days" without a spin up period, but perhaps there was one. In general the modelling details were sparse and need to be filled in a bit.

- Agreed – we felt it appropriate to limit the detail in this paper that is motivated by the coral bleaching and likely to be of most interest to biologists. Further papers have been submitted that consider, in isolation, the modelling and we wished to avoid confusing readers with more detail than useful. We have nonetheless added some more fundamental details as requested to the text.

I was likewise a bit confused about the model/data comparison. Yes, this is difficult to do quantitatively, but no attempt has been made here to do so. Instead it is asserted that the model captures the internal wave variability, with a nod to the figures that show wiggly lines from the data and wiggly lines from the model. (I note here that there is –presumably– a typo: these would be depth-AVERAGED velocities, not depth-integrated velocities...

- Thank you – yes they are depth averaged.

or they are plotted in the wrong units). Comparing depth averaged velocities between a model and data is a bit of a funny thing to do if you care about the baroclinic internal wave response, which is not a depth-averaged sort of quantity. So yeah... I'm not sure outside of qualitatively for the depth averaged currents that there is any evidence that the model is recreating the internal wave features in the data.

- The depth averaged, i.e. barotropic, currents are those used to force the model and this is what is being verified in Figure 8 rather than the baroclinic response; as TPX07 predictions are subject to uncertainty in regions of such bathymetric complexity, we need to be careful in determining whether the model forcing is accurate, thus the value of comparing the amplitudes of predicted and observed tidal velocities.

In S6 both model and a short snapshot of ADCP data in depth and time are shown. But I am confused why the data is only shown for a 2.5 day period, but the full 15 d model time series is shown (see previous comment about fig 5 c and d in the main part of the paper).

- As stated above, we were lucky enough that a mooring was deployed over a slope in November 2019 that was steeper than expected (it was before the multibeam survey!) and as a result slipped down the slope to ~130 m. The benefit for us was that we obtained 2.5 days of data from below the thermocline before the emergency recovery; these are the data used to compare with the model output for November 2019 and are the only observations available from those depths where the bleaching was observed and which we can compare with the model for those depths.

6) Regarding the "mode-2" structure of the tidal variability at MA.

a) First comment is that mode-2 variability is commonly observed and often studied and written about in the literature, so I would remove the statement “relatively rare observation throughout the ocean where attention has largely focused on Mode 1 waves within which the direction of vertical movement of isotherms is consistent throughout the whole water column.” which while perhaps not strictly incorrect, requires a lot of work by the modifiers “relatively” and “largely.”

- OK, although we retain words to the effect that Mode 1 receives more attention which we trust the reviewer accepts.

b) You have the data to show whether or not MA and IDR have a different vertical structure of the internal tide. You **do not** need the model for this. Instead you would use the model to diagnose WHY the internal wave structure is different in the different locations. I would suggest an EOF or other modal decomposition to show this from the data.

- The modal decomposition was performed for the model rather than data because of the lack of useful data during November 2019 at depths where the bleaching was observed. We agree that such an approach to the data analysis would be preferred but given the lack of data at 100 m in MA during November 2019 when the bleaching occurred, we would prefer to avoid further complicating the technical aspects of the oceanography in this biologically-motivated paper.

c) For the record... internal waves cannot “warm” the water. They can transport heat from one place to another (advective heat flux), and they can give up energy to turbulence, which in turn can lead to mixing, which, then, in the presence of a temperature gradient, can lead to a heat flux (mixing-driven heat flux). What you are showing in the model is that the (I think time mean) vertical temperature profile at MA is different from the initial condition (and IDR) particularly in the mid-reef. You do appear to have some data in fig 4 that shows one example of isotherms separating. I would suggest you lean more on these observations. I do NOT think this paper needs to “discover” that the internal wave climate is complicated over complicated bathymetry, or that the oceanography can be different on different sides of an atoll, even a small one, in order to be an impactful contribution.

- We completely agree and thank the reviewer for pointing out that our language may not have been as accurate as it should have been – we have addressed this in the text.
- We hope that the reviewer recognises that we are not stressing our discovery of the complexity of internal waves in such environments but rather demonstrating their potential (or likely) role in driving a response in the mesophotic coral ecosystem.

Minor comments:

1) Lines 182-183: The discussion of SST being a “poor predictor” of thermocline depth and thus the potential that CRW not being good for mesophotic zone corals needs to be more carefully made... the index the authors show in figure 3 b is, in fact, the difference in SSTs between the western and equatorial Indian Ocean. The authors assert that the IOD index captures the anomalous thermocline deepening (which is due most likely to a westward propagating Rossby wave). Nevertheless, after saying that SST as represented in CRW obscures risk to deep corals, in the very next thought they use an SST index as predictive of stress to deep corals. This should be cleaned up.

- This is perhaps a detail that could be debated but ultimately becomes rather technical, and thus distracting given the intended audience, insofar as yes, the IOD index is based on SST values but is quite different to the application of those SST values to a determination of likelihood of shallow coral bleaching. The IOD index simply serves as a metric of how large

the IOD influence may be; the subsequent response is subsurface and not in itself related to the SST that serves of an indicator of the basin-scale state.

2) Lines 221-222: *“resembling at times turbulent bores and, at other times, nonlinear wave trains”*
Where is the evidence for this? This is speculation and should be removed, it doesn't add anything.

- This is one the challenges we experienced when writing this paper – how many examples to use and for what purpose? We have multiple examples of the form taken by the waves (more or less every tidal cycle during 2020 for a start) but can't present them all, despite their interest to physical oceanographers. So, whilst we agree that we'd like to include evidence for this statement in the form of figures, we cannot for practical purposes in this paper.

REVIEWERS' COMMENTS

Reviewer #1 (Remarks to the Author):

The authors do have a nice piece of work here. It is flawed in some ways as all studies are, and has been discussed in several iterations of the review process. I have had my say in this regard and now leave it in the hands of the editors. One final suggestion;

Lines 429-431; suggest it reads as "As the images did not include a uniform Lambertian reference material, a combination of methods was used to assign colonies to these 4 categories:"

Reviewer #3 (Remarks to the Author):

I have reviewed the response of the authors to my initial comments regarding the physical oceanography in the paper. I have also reviewed the revised manuscript and SI. The authors fixed some of the minor things that I pointed out, but largely rebutted the major comments, arguing that their analysis and presentation was sufficiently strong.

I continue to find the presentation of the PO analysis unnecessarily complicated and in places unclear. This is not (and should not be) a paper about the oceanographic details of the internal wave climate in the vicinity of the study site. The analysis and the modelling effort (and probably the data) are not comprehensive enough to perform a detailed study of the dynamics. Of course, a regional study like that would not likely be broadly interesting enough for Nature Communications even if it was comprehensive.

So the authors are in a tough spot. They are trying to make a limited statement about the physical dynamics relative to 1) how out of the ordinary 2019 was and 2) the comparison of the two study sites. The assertion that 2019 was an anomalous year in the central equatorial Indian Ocean oceanography is not novel. Internal waves being important for coral reef thermal stress is not novel. Pointing out internal wave heat flux can differ between adjacent areas in regions of complex bathymetry is also not novel.

I believe the oceanography here can and should be boiled down to two statements: 1) 2019 was regionally an anomalous year with widespread potential for thermal stress even at mesophotic depths,

and that 2) even in the context of regionally high potential for stress, the exposure to internal waves matters on a sub-atoll scale.

To me, this simple case can be made in one or at most 2 figures. I do not believe it matters or is even really that well justified that the internal waves have more of a “mode-2” character. For example, I do not find that having detailed plots of velocity from the models with qualitative comparison to the little data they have is necessary to make the case that they need to make, I find the plan view of the time mean model bottom temperature anomaly relative to the initial condition to be a strange metric for internal wave exposure, etc.

In short there is too much detail and speculation for a paper that is mainly about something else (coral bleaching), and too little for a scientifically defensible study of the oceanography at the sub-atoll scale (a totally different study). It feels as though the compromise the authors have taken with the level of detail leaves the paper in an awkward position. I wish I could be more positive but I think the following statement from the authors sums up my problem:

“We agree with the reviewer that these environments are extremely complicated and other factors may play a role; however, given the challenge of obtaining observation data from this environment and the need to explain the observed bleaching characteristics, we have prioritised the most plausible mechanism for which we have observational and modelling evidence.” (emphasis mine)

If the aim is to provide “plausible mechanisms” because of a “need to explain” – well I would have thought a much more limited presentation of the oceanography would suffice. I disagree with the author’s language here – “needing” to explain something is not hypothesis driven science – but that alone isn’t a deal killer for me.

I thought I had provided perhaps a somewhat over-caffeinated but pretty clear pathway forward for the oceanography part of this paper. It bums me out that I can’t really strongly support this aspect of this study at this stage. I think the paper is important and probably the main finding and presentation is worthy of a high impact journal. It isn’t far away... but right now the oceanography is holding the rest of the story back.

REVIEWERS' COMMENTS

Reviewer #1 (Remarks to the Author):

The authors do have a nice piece of work here. It is flawed in some ways as all studies are, and has been discussed in several iterations of the review process. I have had my say in this regard and now leave it in the hands of the editors. One final suggestion;

Lines 429-431; suggest it reads as "As the images did not include a uniform Lambertian reference material, a combination of methods was used to assign colonies to these 4 categories:"

The lines have been changed accordingly.

Reviewer #3 (Remarks to the Author):

I have reviewed the response of the authors to my initial comments regarding the physical oceanography in the paper. I have also reviewed the revised manuscript and SI. The authors fixed some of the minor things that I pointed out, but largely rebutted the major comments, arguing that their analysis and presentation was sufficiently strong.

We sincerely appreciate the time taken to review the paper again and recognise that the reviewer is frustrated that we largely chose to retain our version of the manuscript. We genuinely endeavoured to address the scientific flaws raised by the reviewer whilst retaining our right as the authors to present the story in the manner that we think appropriate. We appreciate that the reviewer would prefer certain elements to be removed but we consider that those elements are a critical component of the message and removing them would be to the significant detriment of the paper. To be completely clear, we firmly believe that it is precisely because of the detailed insight into the oceanography as a means of explaining why the coral bleached, and why it didn't elsewhere, that makes this paper so important.

I continue to find the presentation of the PO analysis unnecessarily complicated and in places unclear. This is not (and should not be) a paper about the oceanographic details of the internal wave climate in the vicinity of the study site. The analysis and the modelling effort (and probably the data) are not comprehensive enough to perform a detailed study of the dynamics. Of course, a regional study like that would not likely be broadly interesting enough for Nature Communications even if it was comprehensive.

Throughout the presentations at conferences over the past 2 years, and based on the other two reviewers who were not oceanography specialists, we are unable to agree that the PO is complicated at all; indeed we have been very careful to present the complex aspects in a manner that will be accessible to non-specialists and have been repeatedly commended for integrating the PO in a way that is comprehensible to a wider audience.

So the authors are in a tough spot. They are trying to make a limited statement about the physical dynamics relative to 1) how out of the ordinary 2019 was and 2) the comparison of the two study sites. The assertion that 2019 was an anomalous year in the central equatorial Indian Ocean oceanography is not novel. Internal waves being important for coral reef thermal stress is not novel. Pointing out internal wave heat flux can differ between adjacent areas in regions of complex bathymetry is also not novel.

We strongly disagree with this statement and are frustrated by the absence of evidence (in the form of references) provided by the reviewer to demonstrate how the elements of our study are not novel. Internal waves have been invoked as a mechanism responsible for relieving thermal stress in **shallow** corals (for which warming surface temperatures are critical) but we would welcome the provision of references identifying the role of internal waves in modulating the **mesophotic** reef, which is a completely different environment. Secondly, the anomalous conditions of 2019 have been identified in a few papers that we ourselves referenced (as it is only 2023 we would be surprised if there was already a substantial body of work on this event already) but to suggest that identifying the thermocline deepening resulting from the IOD event as the principal driver of coral bleaching in the mesophotic reef is nothing new seems very disingenuous.

I believe the oceanography here can and should be boiled down to two statements: 1) 2019 was regionally an anomalous year with widespread potential for thermal stress even at mesophotic depths, and that 2) even in the context of regionally high potential for stress, the exposure to internal waves matters on a sub-atoll scale.

This is the reviewer's view on how the paper could be written and they are entitled to their opinion but it is equally our right as the authors to retain our version. We stress that the reviewer is not suggesting that the science is flawed but is instead making editorial suggestions that have not been suggested by anyone else during the presentation of this work.

To me, this simple case can be made in one or at most 2 figures. I do not believe it matters or is even really that well justified that the internal waves have more of a "mode-2" character. For example, I do not find that having detailed plots of velocity from the models with qualitative comparison to the little data they have is necessary to make the case that they need to make, I find the plan view of the time mean model bottom temperature anomaly relative to the initial condition to be a strange metric for internal wave exposure, etc.

Again, this is the reviewer's opinion and we disagree. Specifically, we consider that it does matter that we are able to identify the role of Mode 2 waves are being important because this will be applicable across tropical regions with comparable stratification. In future studies during which biologists may work with oceanographers to evaluate how internal waves may alleviate thermal stress, it is obviously enormously helpful for this specific dynamic process to be targeted from the outset as will hopefully be achieved through the reading of this paper. If we withheld this information, for no scientific reason, we would be

holding back precisely the kind of helpful information that scientific papers are published to promote throughout the community. Regarding the plan view, we struggle to understand how this is strange; it demonstrates how the depth bands within which coral bleaching was observed correspond to those depths subjected to elevated temperatures due to the internal waves. We have not heard of any difficulty in understanding this figure and result from any other member of an audience and genuinely consider that the figure is quite easy to understand.

In short there is too much detail and speculation for a paper that is mainly about something else (coral bleaching), and too little for a scientifically defensible study of the oceanography at the sub-atoll scale (a totally different study).

The entire reason this paper is submitted to Nature/Nature Communications is because of the insight into **why** the coral bleached as a response to the oceanographic conditions. We consider, as do everyone else to whom the results have been presented, that the interpretation is robust so we would appreciate what aspect of the study then reviewer considers to be flawed. Similarly, we consider that the level of detail is appropriate and would consider that the support from the first two reviewers (who were specialists in the coral) for the oceanography attests to this.

It feels as though the compromise the authors have taken with the level of detail leaves the paper in an awkward position. I wish I could be more positive but I think the following statement from the authors sums up my problem:

*“We agree with the reviewer that these environments are extremely complicated and other factors may play a role; however, given the challenge of obtaining observation data from this environment and **the need to explain the observed bleaching characteristics**, we have prioritised the most plausible mechanism for which we have observational and modelling evidence.” (emphasis mine)*

If the aim is to provide “plausible mechanisms” because of a “need to explain” – well I would have thought a much more limited presentation of the oceanography would suffice. I disagree with the author’s language here – “needing” to explain something is not hypothesis driven science – but that alone isn’t a deal killer for me.

This is becoming a philosophical debate that we are unable to respond to in a critical, objective manner without the clear identification of what aspects the reviewer would suggest be removed and why.

I thought I had provided perhaps a somewhat over-caffeinated but pretty clear pathway forward for the oceanography part of this paper.

We genuinely appreciate the time taken to provide the review but the recommendations were based on the reviewers vision for what the paper may look like, and they appear to be annoyed that we didn’t make major revisions based on their editorial opinion rather than addressing any scientific flaws. Again, we appreciate the reviewers suggestions but in the

absence of specific points that relate to scientific weakness and/or flaws, we have found it challenging to identify elements that could be changed to strengthen, rather than weaken, the paper.

It bums me out that I can't really strongly support this aspect of this study at this stage. I think the paper is important and probably the main finding and presentation is worthy of a high impact journal. It isn't far away... but right now the oceanography is holding the rest of the story back.

This is an opinion that hasn't been expressed by any other individuals yet and we simply disagree that the PO is anything other than 'excellent' as indicated by the initial reviewers.